# Does Depth Really Hurt GNNs? Injective Message Passing Enables Deep Graph Learning

## Abstract

Graph Neural Networks (GNNs) have shown great promise across domains, yet their performance often degrades with increased depth, commonly attributed to the oversmoothing phenomenon. This has led to a prevailing belief that depth inherently hurts GNNs. In this paper, we challenge this view and argue that the root cause is not depth itself, but the lack of *injectivity* in standard message passing (MP) mechanisms, which fail to preserve structural information across layers. To address this matter, we propose a new message passing layer that is provably injective without requiring any training and guarantees that GNNs match the expressive power of the Weisfeiler-Lehman (WL) test by *design*. Furthermore, this injective MP enables a decoupled GNN architecture where a shallow stack of injective MP layers ensures structural expressivity, followed by a deep stack of feature learning layers for rich representation learning. We provide theoretical analysis on the required depth, width, and initialization of MP layers to ensure both expressivity and numerical stability. Empirically, we demonstrate that our architecture enables deeper GNNs without suffering from oversmoothing. Our findings suggest that depth is not the core limitation in GNNs—lack of injectivity is—and offer a new perspective on building deeper and more expressive GNNs.

## 1 Introduction

In recent years, Graph Neural Networks (GNNs) have emerged as a powerful framework for learning from relational data, achieving state-of-the-art results across a wide range of domains, including molecular property prediction [32, 52, 45], social network analysis [5, 17], and recommendation systems [4, 48, 11]. At their core, GNNs employ a message passing paradigm [14], where node representations are iteratively updated by aggregating information from their neighbors. This structure-aware design enables GNNs to capture both feature and topological patterns in graph-structured data. Despite their success, most GNNs face two fundamental limitations that constrain their scalability and effectiveness: (1) *unreliable expressivity*, as distinguishing graph structures requires injective aggregation, yet standard message passing provides no guarantee of injectivity during training; and (2) *depth-related degradation*, where deeper architectures suffer from oversmoothing, causing node representations to become indistinguishable across the graph.

The expressive power of GNNs is characterized by their ability to distinguish whether two graphs are topologically identical, a problem closely related to the graph isomorphism problem in graph theory, for which no known polynomial-time solution has been found yet [1]. A foundational study by [50] established a close connection between GNNs and the first-order Weisfeiler-Lehman (1-WL) test [30], a widely used graph isomorphism heuristic that distinguishes many non-isomorphic graphs [2]. Their theoretical results show that GNNs can match the discriminative capacity of the WL test if their aggregation scheme is injective. Based on this, Graph Isomorphism Network (GIN) was proposed

to use Multilayer Perceptrons (MLPs) to approximate such injective mappings, making GIN and its variants theoretically as expressive as the 1-WL test.

However, this expressivity is only guaranteed under the assumption that the MLPs remain injective during training. In practice, *GNNs including GIN are trained to minimize task-specific objectives (e.g., node classification), not to enforce injectivity*. As a result, the task-driven optimization of MLPs offers no guarantee that the injective properties required for theoretical expressivity are preserved. This limitation also extends to recent enhancements such as higher-order WL architectures [35] and structural feature augmentation [51, 37], which continue to rely on unverified injectivity assumptions during training. Thus, whether a trained GNN can reliably maintain expressive power equivalent to the 1-WL test (or beyond) throughout learning remains an *open* question.

Beyond expressivity, another long-standing challenge in GNNs is their unusual depth sensitivity. Empirical studies have shown that stacking more message passing layers often leads to degraded performance, a phenomenon widely known as *oversmoothing* [28, 13, 31, 36, 49], where node representations become indistinguishable across the graph. This limits the ability of GNNs to capture long-range dependencies or refine complex representations through deeper architectures. As a result, whereas depth has been considered crucial for the success of deep learning in many fields such as computer vision [20] and natural language processing [42], most GNNs used in practice remains shallow, typically using a fixed depth of just 3-5 layers [31]. While this issue has traditionally been viewed as a depth-related limitation [36, 49], we argue that oversmoothing is fundamentally a symptom of *non-injective propagation*. When message passing functions fail to distinguish structurally distinct neighborhoods, deeper layers only reinforce this loss of information, propagating homogenized features rather than preserving meaningful variations. In this light, we contend that depth is not inherently problematic. Instead, it is the lack of structure-preserving message passing that causes performance collapse as networks grow deeper.

To address these dual challenges, we propose a decoupled GNN architecture that guarantees injective message passing while enabling stable and scalable deep learning over graphs. Our framework combines: (1) a theoretically grounded message passing scheme that is provably injective without training, ensuring WL-level expressivity by design; and (2) a depth-decoupled architecture that separates structure propagation from representation learning, thereby avoiding information homogenization in deep layers, so preventing oversmoothing. Our key contributions are as follows:

- **Injective Message Passing without Training.** We introduce a new message passing layer that leverages the distinct node features in a graph. We prove that simple linear propagation is injective if the distinct node features are linearly independent. To satisfy this condition, we apply a nonlinear feature lifting function to map input features to a linearly independent space, yielding injective propagation without requiring any training.

- **Topology-Aware Depth Selection.** Inspired by the connection between GNN expressivity and the WL test, we establish a principled way to select the number of message passing layers based on graph topology. For example, in social networks with tree-like local structures, we suggest using $\mathcal{O}(\log n)$ message passing layers, as WL iteration typically stabilizes in logarithmic rounds.

- **Decoupled Architecture for Deep Representation Learning.** We identify that existing GNNs entangle structure propagation and representation learning in each layer, which limits expressivity and depth scalability. Hence, we suggest decouple these roles: a small number of injective message passing layers ensure structural expressivity, followed by a deep stack of feature learning layers dedicated to representation refinement.

- **Comprehensive Empirical Validation.** We conduct extensive experiments to validate our theoretical claims and provide detailed ablation studies analyzing the effect of key architectural choices, including network width, number of message passing layers, and number of feature learning layers.

## 2   Related Work

**Expressivity of GNNs**   Since the foundational work by [50], which links GNN expressivity to the 1-WL test, many methods have attempted to improve GNN expressivity by leveraging higher-order architectures [35], positional encodings [51], or random features [37]. While these approaches are theoretically promising, they rely on MLPs trained via task-specific objectives, offering no guarantee

89 of maintaining injectivity throughout learning. In contrast, our method provides provably injective
90 message passing without requiring any training, ensuring expressivity by design.

91 **Oversmoothing and GNN Depth.** Oversmoothing has been studied as a key limitation of deep
92 GNNs [36, 49]. However, prior theoretical work yields conflicting claims—some suggest oversmooth-
93 ing is inevitable on random graphs [36, 49], while others (e.g., via NTK or NNGP analysis) argue it
94 is architecture-dependent [10]. Our view is orthogonal: we identify non-injective propagation as the
95 root cause of oversmoothing. Without injectivity, deeper GNNs propagate homogenized features,
96 leading to representation collapse.

97 **Decoupled Architectures.** Several works have explored decoupling message passing and feature
98 learning, such as SGC [47] and APPNP [29], which simplify or linearize propagation. However,
99 these models still rely on non-injective MP, so their expressivity is not theoretically guaranteed. Our
100 work integrates a provably injective MP layer with a decoupled architecture to ensure both structural
101 expressivity and scalable deep representation learning.

## 3 Preliminaries

103 We begin by reviewing the message passing framework for GNNs and its connection to the expressive
104 power of the 1-dimensional Weisfeiler-Lehman (1-WL) test and setups for theoretical analysis.

105 Let $G = (V, E)$ be an undirected graph, where $V$ is the set of nodes with $|V| = n$ and $E \subseteq V \times V$ is
106 the set of edges. Each node $v \in V$ is associated with a feature vector $\boldsymbol{x}_v \in \mathbb{R}^d$, and let $\boldsymbol{A} \in \{0, 1\}^{n \times n}$
107 denote the adjacency matrix of $G$, where $\boldsymbol{A}_{uv} = 1$ if $(u, v) \in E$. Let $\mathcal{N}(v)$ denote the set of neighbors
108 of node $v$.

109 **Graph Neural Networks.** Modern GNNs [28, 18, 43, 50] iteratively update node embeddings
110 by exchanging and aggregating information over graph neighborhoods. After $k$ layers of message
111 passing, a node's representation captures information from its $k$-hop neighborhood. A general form
112 of message passing GNNs is:

$$\mathbf{m}_v^{(\ell)} = \text{AGGREGATE}^{(\ell)} \left( \{\!\{ \mathbf{h}_u^{(\ell-1)} : u \in \mathcal{N}(v) \}\!\} \right), \quad \mathbf{h}_v^{(\ell)} = \text{COMBINE}^{(\ell)} \left( \mathbf{h}_v^{(\ell-1)}, \mathbf{m}_v^{(\ell)} \right), \quad (1)$$

113 where $\boldsymbol{h}_v^{(\ell)} \in \mathbb{R}^m$ is the embedding of node $v$ at layer $\ell$, initialized as $\boldsymbol{h}_v^{(0)} = \boldsymbol{x}_v$. We assume the
114 embedding dimension to $m$ across layers, simplifying theoretical analysis. The neighborhood is
115 treated as a multiset $\{\!\{\cdot\}\!\}$ to preserve duplicate node information. As shown by [50], a message passing
116 GNN can match the expressive power of the 1-WL test if both the AGGREGATE and COMBINE
117 functions are injective over multisets.

118 **Weisfeiler-Lehman Test.** The graph isomorphism problem asks whether two graphs are structurally
119 identical. Although no known polynomial-time algorithm exists [1], the Weisfeiler-Lehman (WL) test
120 [30] is a widely used heuristic that distinguishes many non-isomorphic graphs [2]. The 1-WL test,
121 also known as color refinement, iteratively updates node labels by aggregating labels from neighbors:

$$c_v^{(\ell)} = \text{HASH} \left( c_v^{(\ell-1)}, \{\!\{ c_u^{(\ell-1)} : u \in \mathcal{N}(v) \}\!\} \right). \quad (2)$$

122 Two graphs are declared non-isomorphic if their label multisets differ at any iteration. The 1-WL test
123 forms the basis for theoretical GNN expressivity analysis [50, 35].

124 **Assumptions for Expressivity Analysis.** Following [50, 35], we assume that both the GNN and
125 the 1-WL test start from identical node features across compared graphs. This allows us to isolate
126 structural discriminability, as any differences must be inferred purely from topology. This setup
127 reflects a worst-case scenario, since identical initialization typically requires more WL iterations than
128 heterogeneous ones. Hence, our analysis in Section 5.1 thus provides an upper bound on the MP
129 depth required for expressivity. Additionally, we assume that the final graph representation is the
130 multiset of node embeddings $\{\!\{ \boldsymbol{h}_v^{(L)} : v \in V \}\!\}$, rather than a pooled vector. This keeps our analysis
131 focused at the node level. For graph-level tasks, a READOUT function (e.g., sum) is often used to
132 obtain a single vector, but unless this function is also injective, it may obscure structural differences
133 [50]. Therefore, injective message passing is a necessary condition for expressivity: once node
134 embeddings collapse, no readout function can recover the lost structural information.

## 4   Training-Free Injective Message Passing

In this section, we present a new message passing (MP) layer that is provably injective without any training. As a result, a GNN using this MP layer is guaranteed to match the expressive power of the 1-WL test. Our design is based on a two-step construction: we first show that a linear MP layer is injective under linearly independent embeddings, and then demonstrate that this condition can be satisfied via a simple nonlinear feature transformation.

### 4.1   Injectivity from Linear Message Passing

Let $S_v^{(\ell)} := \{\!\{ \boldsymbol{h}_u^{(\ell)} : u \in \mathcal{N}(v) \}\!\}$ denote the multiset of neighbor embeddings for node $v$ at layer $\ell$. We consider the following linear message passing scheme:

$$\boldsymbol{h}_v^{(\ell)} = (1 + \epsilon)\boldsymbol{h}_v^{(\ell-1)} + \sum_{u \in \mathcal{N}(v)} \boldsymbol{h}_u^{(\ell-1)}, \tag{3}$$

where $\epsilon$ is a fixed irrational number[1]. The effectiveness of such linear aggregation has been observed in prior work. For instance, [47] shows that purely linear message passing performs competitively across various tasks. Moreover, this form was also used in the expressivity proof of GIN [50], where node features are encoded as $n$-digit scalars, making the message passing injective. Inspired by these insights, we generalize the idea and show that linear message passing is injective whenever the set of distinct node embeddings is linearly independent.

**Proposition 1.** *Suppose all distinct node embeddings $\{\boldsymbol{h}_v^{(\ell)}\}_{v \in V}$ are linearly independent, and $\epsilon$ is irrational. Then the linear message passing update equation 3 is injective over all neighborhood multisets $S_v^{(\ell)}$.*

Compared to prior expressivity analysis that assumes bounded multiset cardinality [50, 25, 35], this condition is weaker and more scalable, especially for large graphs. However, the linear independence assumption can easily be violated as message passing layers accumulate more diverse embeddings. To address this, we introduce a nonlinear transformation that lifts features into a linearly independent space.

### 4.2   Guaranteeing Linear Independence via Nonlinearity

While Proposition 1 shows that linear message passing is injective under linearly independent node embeddings, this condition is fragile in practice, especially as deeper GNN layers accumulate more structurally distinct embeddings. To address this, we propose lifting node embeddings into a linearly independent space using a simple nonlinear transformation.

Specifically, we apply a one-layer MLP of the following form:

$$\boldsymbol{a}_v = \frac{1}{\sqrt{m}}\phi(\boldsymbol{W}^\top \boldsymbol{h}_v), \quad \forall v \in V, \tag{4}$$

where $\boldsymbol{W} \in \mathbb{R}^{d \times m}$ is a weight matrix, and $\phi$ is a nonlinear activation function. To stabilize the output magnitudes, we apply the standard scaling factor $\frac{1}{\sqrt{m}}$ for standard random initialization schemes on the weights matrix $\boldsymbol{W}$ such as He or Xavier initialization [19, 15]:

$$\boldsymbol{W}_{ij} \overset{\text{i.i.d.}}{\sim} \mathcal{N}(0, \sigma^2). \tag{5}$$

Let $\boldsymbol{H} \in \mathbb{R}^{k \times d}$ be a matrix composed of all $k$ distinct input embeddings. Define the lifted feature matrix as:

$$\boldsymbol{A} = \frac{1}{\sqrt{m}}\phi(\boldsymbol{H}\boldsymbol{W}) \in \mathbb{R}^{k \times m}. \tag{6}$$

We aim to show that with high probability over the random initialization of $\boldsymbol{W}$, the rows of $\boldsymbol{A}$ are linearly independent if $m$ is near-linear in $k$. Our key result is summarized as follows, and the proof, which builds on the notion of dual activation function [7] and concentration bounds from random matrix theory [44, 3], is deferred to the appendix.

---

[1]In [50], the authors evaluate both learned and fixed values of $\epsilon$ and find no significant difference in empirical performance.

**Proposition 2.** *Let $\phi$ be Lipschitz continuous, nonlinear, and non-polynomial. Let $\boldsymbol{H} \in \mathbb{R}^{k \times d}$ contain $k$ distinct input embeddings and $\boldsymbol{A} \in \mathbb{R}^{k \times m}$ be the lifted feature matrix. Then, for any $\delta > 0$, if $m = \Omega\left(\frac{k}{\lambda_0} \log\left(\frac{k}{\lambda_0}\right) \log\left(\frac{k}{\delta}\right)\right)$, then with probability at least $1 - \delta$, the matrix $\boldsymbol{A}$ has linearly independent rows, where $\lambda_0 := \lambda_{\min}\left(\mathbb{E}\left[\sigma(\boldsymbol{H}\boldsymbol{w})\sigma(\boldsymbol{H}\boldsymbol{w})^{\top}\right]\right) > 0$.*

Proposition 2 shows that linearly independent node embeddings can be achieved via a simple one-layer nonlinear transformation. As the weights $\boldsymbol{W}$ are fixed at initialization, no training is required to ensure injectivity, offering a training-free guarantee, differing fundamentally from prior MLP-based message passing schemes like GIN and its variants [50, 35]. Remarkably, the required width $m$ scales only near-linearly with the number of distinct embeddings $k$, *i.e.*, $m = \tilde{\Omega}(k)$, rather than the total number of nodes $n$, as used in prior analysis [35, 50, 25], making this approach efficient and *scalable* for large-scale graph representation learning in practice.

### 4.3 WL-Expressive GNN via Injective Message Passing

By combining Proposition 1 and Proposition 2, we can construct a training-free message passing layer that is provably injective:

$$\boldsymbol{a}_v^{(\ell)} = \frac{1}{\sqrt{m}}\phi(\boldsymbol{W}^{(\ell)}\boldsymbol{h}_v^{(\ell-1)}), \tag{7}$$

$$\boldsymbol{h}_v^{(\ell)} = (1 + \epsilon)\boldsymbol{a}_v^{(\ell)} + \sum_{u \in \mathcal{N}(v)} \boldsymbol{a}_u^{(\ell)} \tag{8}$$

where $\boldsymbol{W}^{(\ell)} \in \mathbb{R}^{m \times m}$ is assumed to have the same width across different layers for simplicity.

By following [50], which shows the expressive power of GNN can match 1-WL test with injective message passing, we obtain the following key results for our proposed new message passing layer.

**Theorem 1.** *Suppose $\phi$ is Lipschitz continuous and nonlinear but non-polynomial, and $\epsilon$ is an irrational number. Let $k$ denote the total number of distinct rooted subtree structures encountered during the $1$-WL test across both graphs. For any $\delta > 0$, if the width $m$ satisfies $m = \Omega\left(\frac{k}{\lambda_0} \log\left(\frac{k}{\lambda_0}\right) \log\left(\frac{k}{\delta}\right)\right)$, then with probability at least $1 - \delta$ over the random initialization, a GNN using the message passing layer in Eq. (8) is as expressive as the $1$-WL test. That is, it produces different embeddings for two graphs if and only if the $1$-WL test distinguishes them.*

**Remark 1.** *While the total number of nodes $n$ provides a loose upper bound on network width $m$, our result shows that the required width $m$ depends only on the number of distinct rooted subtrees $k$ observed during 1-WL iterations. In many practical datasets, $k \ll n$, leading to a significantly smaller width requirement. This reflects the fact that 1-WL distinguishability depends on structural diversity rather than graph size.*

Since expressivity is already ensured by design, the focus of training can now shift mainly toward representation learning, just as in classical deep networks. This motivates a clean architectural separation between structure propagation and representation learning. In the next section, we introduce a decoupled GNN design that leverages this separation to enable deeper and more stable feature learning while maintaining expressive power.

## 5  Layer Design: Decoupling Message Passing and Feature Learning

Most existing GNNs adopt a shallow and fixed number of message passing (MP) layers (e.g., 3–5) [31, 36, 49], as performance often degrades with increased depth due to oversmoothing. Our key insight is that depth itself is not the root cause of this degradation; rather, it stems from non-injective aggregation schemes, which fail to preserve structural variations and instead propagate homogenized features. In contrast, the training-free injective MP scheme introduced in Section 4 guarantees expressivity by design. Once structural information is sufficiently captured through a small number of such injective MP layers, the model can shift its focus to representation learning. This motivates a decoupled architecture: a shallow but expressive MP block for structural encoding, followed by a deep and fully trainable feature learning (FL) block for downstream tasks.

## 5.1 Topology-Aware Message Passing Depth

We first examine how many MP layers are required to sufficiently explore structural variations in a graph. Inspired by the WL test, we argue that the number of MP layers should not be fixed across datasets, but instead determined by the graph's topology. Specifically, the depth should align with the number of WL iterations needed to distinguish node labels.

While the 1-WL test stabilizes in just $\mathcal{O}(1)$ iterations for fully connected graphs, the worst-case complexity is $\mathcal{O}(n)$, as the number of distinct node labels may grow linearly. Recent theoretical results confirm this bound is tight: Kiefer and McKay [26] construct an infinite family of graphs requiring $n$ iterations, and Grohe et al. [16] further characterize families with high iteration counts. Motivated by this, we state the following general upper bound:

**Proposition 3.** *For two $n$-node graphs with identical initial node features, an expressive GNN with $\mathcal{O}(n)$ injective MP layers returns different embeddings if and only if the $1$-WL test determines them non-isomorphic.*

While informative, this worst-case bound is rarely encountered in practice. Most real-world graphs exhibit structural regularities that allow for sublinear MP depths, as shown in the following proposition:

**Proposition 4.** *For two $n$-node graphs with identical initial node features, suppose the $1$-WL test determines them non-isomorphic. Then:*

- *If the graphs are balanced binary trees, then $\mathcal{O}(\log n)$ injective MP layers suffice.*

- *If the graphs are 2D grids, then $\mathcal{O}(\sqrt{n})$ injective MP layers suffice.*

**Remark 2.** *Social networks often exhibit small-world properties, where the graph diameter is much smaller than $n$. Local structures tend to be either tree-like (e.g., with hub nodes) [24], where $\mathcal{O}(\log n)$ layers are sufficient, or grid-like with uniform connectivity [46], where $\mathcal{O}(\sqrt{n})$ layers are more appropriate.*

**Remark 3.** *Molecular graphs generally have bounded degree and a small number of atom types, resulting in low-diameter and often planar structures [14, 9]. We recommend using $\mathcal{O}(\log n)$ MP layers for such datasets.*

These observations highlight the importance of *topology-aware* MP depth selection. Rather than using a fixed shallow depth, practitioners can estimate graph diameter or WL iteration count via breadth-first search (BFS) or symbolic labeling to guide the number of MP layers.

## 5.2 Stability of Deep Message Passing

Although many real-world graphs only require shallow MP depth, the worst-case $\mathcal{O}(n)$ bound motivates analyzing stability when stacking many injective MP layers. Since our MP is training-free, the variance of hidden representations is fully governed by initialization. Let $\boldsymbol{H}^{(\ell)}$ denote the node embedding matrix at layer $\ell$. We derive the following recursive bound on its norm.

**Lemma 1.** *If $\phi$ is $L$-Lipschitz, then*

$$\mathbb{E}\left[\|\boldsymbol{H}^{(\ell)}\|^2\right] \leq \left\{\sigma_\ell L\left[(1+\epsilon)+\|\boldsymbol{A}\|\right]\right\}^2 \mathbb{E}\left[\|\boldsymbol{H}^{(\ell-1)}\|^2\right], \tag{9}$$

*where $\boldsymbol{A} \in \{0,1\}^{n \times n}$ is the adjacency matrix.*

To maintain stable propagation, we recommend initializing MP layers with:

$$\sigma_\ell = \frac{1}{L\left[(1+\epsilon)+\|\boldsymbol{A}\|\right]}. \tag{10}$$

Although our architecture does not require training the MP layers to ensure expressivity, we empirically observe that training the ML layers can further improve performance (see Figure 1 and 2). Moreover, even when MP layers are trained, this initialization remains beneficial, since parameters typically remain close to their initial values during training [23].

### 5.3 Decoupled GNN Design: Injective Propagation + Deep Learning

Standard GNNs couple MP and FL within each layer. This entanglement introduces two key limitations. First, MP functions are optimized indirectly via task-specific objectives, which—as discussed in Section 4—cannot guarantee injectivity and thus undermine expressive power. Second, as non-injective MP layers accumulate, they tend to propagate homogenized features rather than preserve structural distinctions, leading to oversmoothing and degraded performance in deep models.

In contrast, deep models in other domains, such as CNNs in vision [20, 21] and Transformers in NLP [42], succeed by separating early structural encoding from deep feature refinement. For example, CNNs resolve local edges and textures in early layers, while deeper layers capture abstract, task-specific features.

We bring this principle to graph learning through a decoupled architecture:

- A shallow stack of injective MP layers to encode structural variation and ensure WL-level expressivity.
- A deep stack of fully trainable FL layers to learn rich representations for downstream tasks, leveraging the expressive embeddings from the MP block.

This separation yields multiple benefits:

- **Guaranteed Expressivity**: The injective MP layers encode sufficient structural variations by design, making the GNN WL-expressive by construction.
- **Focused Training**: Since expressivity is ensured upfront, all optimization effort is devoted to learning useful representations, not propagation.
- **Deeper and Stable Architectures**: The feature learning block can incorporate standard stabilization and regularization techniques (e.g., residual connections [20], batch normalization [22], and dropout [41]), enabling significantly deeper GNNs than previously feasible.
- **Scalability**: As established in Section 5.1, the depth and width of MP layers only need to scale with the number of distinct rooted subtree structures $k$ and graph diameter $D$, respectively, both of which are much smaller than the total number of nodes $n$ in most real-world datasets, while the FL block can be made much deeper.

Models like APPNP [29] and SGC [47] also decouple structure and learning, but use non-injective linear MPs. Consequently, they still suffer from oversmoothing and under-expressivity. In contrast, our decoupled design pairs injective propagation with deep learning, enabling both expressivity and scalability.

## 6 Experiments

Table 1: Node classification results over eight datasets (%). The best results are highlighted in blue.

|         | Cora | Citeseer | Pubmed | Wikics | Computer | Physics | CS | Photo |
|---------|------|----------|--------|--------|----------|---------|-----|-------|
| # Nodes | 2,708 | 3,327 | 19,717 | 11701 | 13,752 | 34,493 | 18,333 | 7,650 |
| # Edges | 5,278 | 4,732 | 44,324 | 216,123 | 245,861 | 247,962 | 81,894 | 119,081 |
| Metric  | Accuracy | Acccuracy | Accuracy | Accuracy | Accuracy | Accuracy | Accuracy | Accuracy |
| GCN | $84.08_{\pm0.37}$ | $72.28_{\pm0.74}$ | $80.48_{\pm0.81}$ | $80.41_{\pm0.57}$ | $93.99_{\pm0.26}$ | $97.38_{\pm0.07}$ | $95.91_{\pm0.05}$ | $95.73_{\pm0.19}$ |
| SAGE | $83.36_{\pm0.34}$ | $71.96_{\pm1.11}$ | $78.34_{\pm0.53}$ | $80.60_{\pm0.18}$ | $93.14_{\pm0.13}$ | $97.19_{\pm0.10}$ | $96.26_{\pm0.07}$ | $96.34_{\pm0.57}$ |
| GAT | $82.92_{\pm1.48}$ | $71.94_{\pm0.99}$ | $80.32_{\pm0.60}$ | $81.02_{\pm0.36}$ | $93.77_{\pm0.13}$ | $97.30_{\pm0.08}$ | $96.18_{\pm0.12}$ | $96.56_{\pm0.39}$ |
| Ours | $85.06_{\pm0.83}$ | $72.19_{\pm0.81}$ | $81.64_{\pm0.63}$ | $81.10_{\pm0.29}$ | $93.99_{\pm0.15}$ | $97.54_{\pm0.08}$ | $96.27_{\pm0.23}$ | $96.21_{\pm0.10}$ |

In this section, we conduct a series of experiments to support our theoretical findings and design principles. We focus our experiments on studying the expressivity of the proposed method through checking the injectivity of message passing and the scalability of the depth to prevent oversmoothing. The exact details about the experiments and choice of datasets are deferred to the appendix due to page limit.

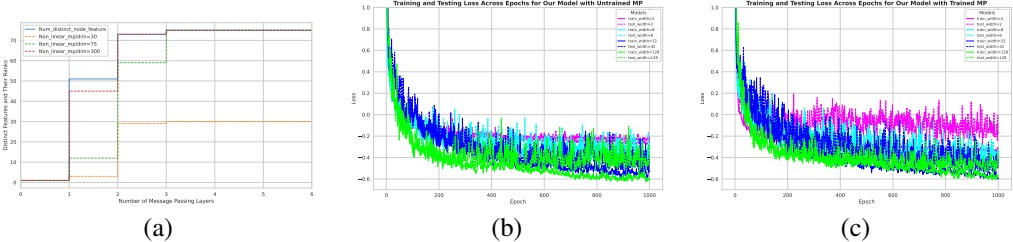

|          |          |          |
| :------: | :------: | :------: |
| (a)      | (b)      | (c)      |

Figure 1: Effect of **MP width** on expressivity and training: (a) Larger widths yield higher feature matrix rank, closely tracking the number of distinct node features; (b) Wider MPs reduce training loss but risk overfitting; (c) Training MP parameters alleviates overfitting.

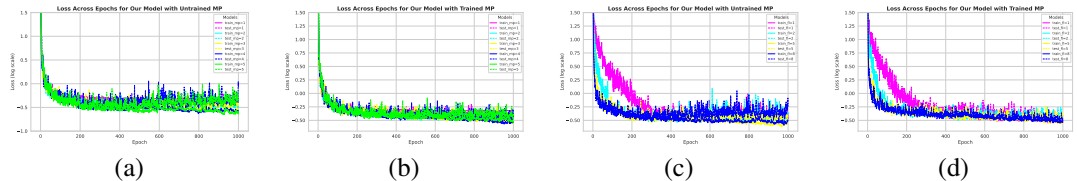

|        |        |        |        |
| :----: | :----: | :----: | :----: |
| (a)    | (b)    | (c)    | (d)    |

Figure 2: Effect of **MP depth** and **FL depth** on learning: (a) On molecular graphs, deeper frozen MPs reduce loss but cause overfitting; (b) Training MP layers mitigates overfitting on molecular graphs; (c) With frozen MP layers, deeper FL improves training loss but risks overfitting; (d) Training MP layers stabilizes learning and improves generalization as FL depth increases.

**Expressive GNNs and Injective Message Passing.** As shown by [50], a GNN achieves 1-WL expressivity if its MPs are injective. In Section 4, we proposed a new MP layer in Eq. (8) that is provably injective without training, based on Propositions 1 and 2. We empirically validate our construction by tracking the number and rank of distinct node features across layers. As shown in Figure 1(a), even with identical initial features, our MP steadily increases both the number of distinct node embeddings and their rank, provided the hidden dimension $m$ exceeds the number of distinct features $k$. Due to numerical instability (e.g., near-zero singular values), the rank may briefly lag when $m \approx k$, but eventually matches it.

Importantly, this injectivity is achieved without training. Moreover, the required hidden width scales with the number of distinct structures, *i.e.*, $m = \tilde{\Omega}(k)$, not the total number of nodes $n$, with $k \ll n$ in practice. For example, in Figure 1(a), a graph with $n = 218$ nodes and $k = 75$ distinct rooted subtree structures reaches full expressivity using $m = 75$ and only 3 MP layers, demonstrating the scalability of our method to large graphs.

**Impact of Message Passing Width.** To assess how MP width affects model performance, we vary the width of our injective MP layer Eq. (8) while keeping its parameters frozen and only training the feature learning layers, following our decoupled design in Section 5. As shown in Figure 1(b), increasing the width consistently lowers training loss, demonstrating that wider MPs can improve learning, even without being trained.

However, excessive width introduces overfitting, likely because randomly initialized MP layers inject noise that is memorized by the downstream feature learner. Figure 1(c) shows that training the MP layers mitigates this issue, especially for large widths. Notably, smaller-width models exhibit unstable performance when *MP and FL are entangled*, encouraging a decoupled GNN design as we introduced in Section 5.3. It also highlights that large-width MPs may not only guarantee injectivity at initialization but also preserve it during training—a promising direction for future investigation.

**Impact of Message Passing Depth.** As discussed in Section 5.1, the number of MP layers should match the number of 1-WL iterations required to distinguish structural patterns. For molecular graphs, which are typically tree-like, $\mathcal{O}(\log n)$ MP layers are sufficient. Figure 2(a) shows that increasing MP depth initially improves training loss, but excessive depth eventually degrades test performance, an effect resembling overfitting. This mirrors our earlier observation with large MP width: frozen, randomly initialized MP layers introduce noise that downstream FL layers may overfit. As shown in

Figure 2(b), allowing MP layers to be trained mitigates this issue and improves both training and test performance. While our injective MP design does not require training for expressivity, these results suggest that training can further enhance performance, possibly because injectivity is preserved throughout optimization.

**Decoupled Deep Representation Learning.** Traditional GNNs entangle structure propagation and representation learning within each layer, limiting the model's capacity to learn expressive representations, especially as deeper architectures often lead to oversmoothing. Our proposed decoupling strategy from Section 5 separates these two components: a small number of injective MP layers ensures structural expressivity, while deep FL layers refine node representations for downstream tasks.

To evaluate this design, we fix the number of MP layers based on the dataset's graph topology, as suggested in Section 5.1, and vary the number of FL layers. As shown in Figure 2(c), increasing the FL depth reduces training loss, but frozen MP layers (initialized randomly) can introduce noise, leading to overfitting for deeper FL models. In Figure 2(b), this overfitting is mitigated when we train the MP layers alongside FL layers, resulting in both stable training and improved generalization.

At first glance, training the MP layers may seem to reintroduce the entanglement of propagation and feature learning seen in traditional GNNs. However, we observe no oversmoothing, even with increased depth. This key difference lies in our use of provably injective MP layers. Unlike prior GNNs where non-injective propagation leads to homogenized features and expressivity loss, our MP layers preserve structural distinctions throughout training. This supports our central claim: depth is not the issue, but non-injective message passing is. By addressing this, our design enables deep, expressive, and stable GNNs.

**Comparison with Classic GNNs.** To further validate the practical effectiveness of our architecture, we compare our model against several state-of-the-art GNNs. Recent studies (e.g., [33]) have shown that classic GNNs can achieve remarkably strong performance. Following their setup, we evaluate our model under comparable conditions. Notably, our message passing depth and width are selected based on theoretical insights from Section 4 and 5, ensuring sufficient structural exploration without excessive overhead. Although our comparison does not involve exhaustive hyperparameter tuning, the results in Table 1 show that our model achieves competitive or even superior performance. We attribute this to two key factors: (1) our injective MP layers preserve expressivity throughout training by design, and (2) the decoupled architecture enables deep and stable feature learning. Together, these components allow our model to scale effectively while maintaining high expressivity, bridging a critical gap between depth, learnability, and structural awareness in GNN design.

# 7 Conclusion

This paper revisits two long-standing challenges in GNNs: unreliable expressivity and depth-induced degradation. While depth is often blamed for oversmoothing, we argue that the true bottleneck lies in non-injective message passing, which causes structural information to vanish across layers. To address this, we propose a provably injective message passing scheme that requires no training and matches the expressive power of the 1-WL test by design. Built on this foundation, we introduce a decoupled GNN architecture that separates structural propagation from feature learning. This design allows shallow, injective MP layers to encode topology and supports arbitrarily deep feature learning blocks for expressive representation learning. We further develop theory-guided criteria for selecting MP depth based on graph topology and propose a variance-stabilized initialization scheme to ensure robustness across depths. Extensive empirical studies validate our claims: the proposed decoupled GNN with injective MPs avoids oversmoothing, remains expressive at scale, and enables deep, stable architectures. By disentangling expressivity from trainability, our framework bridges a critical gap in GNN design, bringing the scalability of deep learning to graph representation learning. We hope this work inspires future exploration into principled, modular GNN architectures that are both expressive and trainable, especially in large-scale or structure-sensitive domains.

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

## A    Mathematic Proofs

## B    Useful Mathematical Results

**Lemma 2** (Matrix Chernoff inequalities). *Consider a finite sequence $\{\boldsymbol{X}_k\}$ of independent, random, self-adjoint matrices with dimension $d$. Assume that each random matrix satisfies $0 \preceq X_k \preceq R I_n$. Denote*

$$\mu_{\min} = \lambda_{\min}\left\{\mathbb{E}\left(\sum_k X_k\right)\right\}, \quad \mu_{\max} = \lambda_{\max}\left\{\mathbb{E}\left(\sum_k X_k\right)\right\}. \tag{11}$$

*Then*

$$\mathbb{P}\left\{\lambda_{\max}\left(\sum_k X_k\right) \geq (1+\delta)\mu_{\max}\right\} \leq n\left[\frac{e^\delta}{(1+\delta)^{1+\delta}}\right]^{\mu_{\max}/R}, \quad \forall \delta \geq 0. \tag{12}$$

$$\mathbb{P}\left\{\lambda_{\min}\left(\sum_k X_k\right) \leq (1-\delta)\mu_{\min}\right\} \leq n\left[\frac{e^{-\delta}}{(1-\delta)^{1-\delta}}\right]^{\mu_{\min}/R}, \quad \forall 0 < \delta < 1. \tag{13}$$

## C    Missing Proofs for Section 4

This section includes the missing proofs in Section 4 that design and prove injectivity of message passing, which is further be leveraged to show the expressive power of GNNs.

### C.1    Linear Injective Message Passing

In this subsection, we prove Proposition 1. Recall that the message-passing layer is defined as follows

$$\boldsymbol{h}_v^\ell = \phi(\boldsymbol{h}_v^{\ell-1}, f(\{\!\{\boldsymbol{h}_u^{\ell-1} : u \in \mathcal{N}_v\}\!\}) \tag{14}$$

where $f$ is the aggregation operation and $\phi$ is the combination operation.

Let us suppose a graph with $n$ vertices, and there are totally $k$ **distinct** features. Additionally, we assume all distinct features are linearly independent. Then we can show both $f$ and $\phi$ are injective by using a simple linear map.

First, we consider the aggregation operation $f$ defined as follows

$$f(\{\!\{\boldsymbol{h}_u^{\ell-1} : u \in \mathcal{N}_v\}\!\}) = \sum_{u \in \mathcal{N}_v} \boldsymbol{h}_u^{\ell-1}. \tag{15}$$

Let $\boldsymbol{H} \in \mathbb{R}^{k \times d}$ be the distinct node features. Then we have

$$f(\{\!\{\boldsymbol{h}_u^{\ell-1} : u \in \mathcal{N}_v\}\!\}) = \boldsymbol{H}^\top \alpha_v, \tag{16}$$

where $\alpha_v$ is an integer vector whose entries are all nonnegative integers and their sum equals the degree of node $v$. If the two nodes have the same aggregated messages, then we have

$$\boldsymbol{H}^\top \alpha_v = \boldsymbol{H}^\top \alpha_u \tag{17}$$

or equivalently

$$\boldsymbol{H}^\top (\alpha_v - \alpha_u) = 0. \tag{18}$$

Recall that we have distinct features that are linearly independent. Hence, $\boldsymbol{H}^\top$ is full column rank and $\alpha_u = \alpha_v$. This discussion proves the following result.

**Lemma 3.** *If distinct node features are linearly independent, then the linear aggregation function $f$ is injective.*

Next, let us recall the linear message passing layer:

$$\boldsymbol{h}_v^\ell = (1+\epsilon)\boldsymbol{h}_v^{\ell-1} + \sum_{u \in \mathcal{N}_v} \boldsymbol{h}_u^{\ell-1} \tag{19}$$

To simplify the notation, we denote $\{\!\{h_v^{\ell-1}\}\!\} := \{\!\{h_u^{\ell-1} : u \in \mathcal{N}_v\}\!\}$. Given two inputs $(h_v^{\ell-1}, \{\!\{h_v^{\ell-1}\}\!\})$ and $(h_{\bar{v}}^{\ell-1}, \{\!\{h_{\bar{v}}^{\ell-1}\}\!\})$, we consider

$$h_v^\ell = h_{\bar{v}}^\ell, \tag{20}$$

or equivalently

$$(1+\epsilon)(h_v^{\ell-1} - h_{\bar{v}}^{\ell-1}) = H_k^\top(\alpha_{\bar{v}} - \alpha_v) \tag{21}$$

If $h_v^{\ell-1} = h_{\bar{v}}^{\ell-1}$, then we have $H^\top(\alpha_{\bar{v}} - \alpha_v) = 0$ implies $\alpha_{\bar{v}} = \alpha_v$ as we have proved before.

Now we will show $h_v^{\ell-1} \neq h_{\bar{v}}^{\ell-1}$ is impossible if we chose $\varepsilon$ as a irrational number. Note that the LHS is the linear span of $h_v^\ell$ and $h_{\bar{v}}^\ell$. The RHS can only be their linear span as they are linearly independent. Hence, there exist nonzero integers $a$ and $b$ such that

$$H^\top(\alpha_{\bar{v}} - \alpha_v) = ah_v^{\ell-1} + bh_{\bar{v}}^{\ell-1}. \tag{22}$$

Additionally, we will have $b = -a$. Otherwise, $h_v$ and $h_{\bar{v}}$ becomes linearly dependent. Then we obtain

$$(1+\epsilon)(h_v^{\ell-1} - h_{\bar{v}}^\ell - 1) = a(h_v^{\ell-1} - h_{\bar{v}}^{\ell-1}) \tag{23}$$

Recall that $a$ is an integer, while $\epsilon$ is irrational. Hence, we must have $h_v = h_{\bar{v}}$.

**Lemma 4.** *Suppose distinct node features are linearly independent. If we choose $\epsilon > 0$ to be irrational, then the linear message passing is injective.*

Combining Lemma 3 and Lemma 4 yields the desired result in Proposition 1.

## C.2 Nonlinear Lifting Enables Linearly Independence

We have shown that linear message passing is injective if distinct node feature vectors are linearly independent. However, this linear message passing cannot be done in an iterative manner since after one linear message passing, the distinct feature vectors become likely linearly dependent. Hence, in this section, we will show how to use a nonlinear transform to lift the distinct feature vectors to become linearly independent.

Let us consider that we have a totally of $k$ distinct feature vectors and they can be linearly dependent or not. For simplicity, we denote them as $X \in \mathbb{R}^{k \times d}$. Then we consider the following nonlinear transform

$$H = \frac{1}{\sqrt{m}}\phi(XW) \in \mathbb{R}^{k \times m}, \tag{24}$$

where $W \in \mathbb{R}^{d \times m}$. This is a one-layer MLP. Moreover, we will assume $\|x\| = 1$ and we random initialize $W$ such that

$$W_{ij} \overset{i.i.d.}{\sim} \mathcal{N}(0,1). \tag{25}$$

Hence, to show $H$ has linearly independent rows, it is equivalent to study the smallest eigenvalues of the following Gram matrix

$$G = HH^\top = \frac{1}{m}\phi(XW)\phi XW)^\top = \frac{1}{m}\sum_{r=1}^m \phi(Xw_r)\phi(Xw_r)^\top, \tag{26}$$

where $w_r \overset{i.i.d.}{\sim} \mathcal{N}(0, I_d)$. With law of large number argument, as $m \to \infty$, we have $G$ converges to

$$G^\infty = \mathbb{E}[\phi(Xw)\phi(Xw)^\top]. \tag{27}$$

Then we can show that as long as $\phi$ is nonlinear but non-polynomial, the least eigenvalue of $K^\infty$ is strictly positive definite.

The proof is based on the notion of dual activation [7]. The determinsitic matrix $G$ has the following expansion form:

$$G^\infty = \mathbb{E}\phi(Xw)\phi(Xw)^\top = \sum_{n=0}^\infty a_n^2 (XX^\top)^{\odot n}, \tag{28}$$

where $a_n$ is the $n$-th Hermitian coefficients of $\phi$ and $\odot$ is the element-wise product. Then it is followed by [12, Lemma A.9] that the least eigenvalue is strictly positive definite.

**Lemma 5.** *Suppose $\boldsymbol{x}_i \in \mathbb{S}^{d-1}$ for all $i \in [n]$. If $\boldsymbol{x}_i \neq \boldsymbol{x}_j$ and $\phi$ is non-polynomial, then $\lambda_0 = \lambda_{\min}(\boldsymbol{G}^\infty) > 0$.*

Remarkably, the matrix $\boldsymbol{G}^\infty$ is deterministic, and it is the limit of $\boldsymbol{G}$ as $m$ tends to infinity. Next, we will show that the least eigenvalue of $\boldsymbol{G}$ is highly likely to be positive even if $m$ is near-linear in $k$.

We have

$$\boldsymbol{A}\boldsymbol{A}^\top = \sum_{r=1}^{m} \phi(\boldsymbol{X}\boldsymbol{w}_r)\phi(\boldsymbol{X}\boldsymbol{w}_r)^\top \tag{29}$$

where we denote $\boldsymbol{A} = \phi(\boldsymbol{X}\boldsymbol{W})$. As $\boldsymbol{w}_r$ are i.i.d., we have

$$\mathbb{E}\boldsymbol{A}\boldsymbol{A}^\top = m\boldsymbol{G}^\infty. \tag{30}$$

And so we obtain

$$\lambda_{\min}(\mathbb{E}\boldsymbol{A}\boldsymbol{A}^\top) = m\lambda_0. \tag{31}$$

For any positive $t > 0$, we consider the matrix $\boldsymbol{B}_t$ with each column defined

$$\boldsymbol{b}_r = \phi(\boldsymbol{X}\boldsymbol{w}_r)1\{\|\phi(\boldsymbol{X}\boldsymbol{w}_r)\| \le t\}. \tag{32}$$

Then we have

$$\lambda_{\max}(\boldsymbol{b}_r\boldsymbol{b}_r^\top) \le t^2 \tag{33}$$

and

$$\lambda_{\min}(\boldsymbol{A}\boldsymbol{A}^\top) \ge \lambda_{\min}(\boldsymbol{B}_t\boldsymbol{B}_t^\top), \quad \forall t > 0. \tag{34}$$

For any $\epsilon > 0$, we apply the Matrix Chernoff inequality to the matrices $\{\boldsymbol{b}_r\boldsymbol{b}_r^\top\}$ and get

$$\mathbb{P}\left\{\lambda_{\min}(\boldsymbol{B}_t\boldsymbol{B}_t^\top) \le (1-\epsilon)\lambda_{\min}(\mathbb{E}\boldsymbol{B}_t\boldsymbol{B}_t^\top)\right\} \le k e^{-\epsilon^2 \lambda_{\min}(\mathbb{E}\boldsymbol{B}_t\boldsymbol{B}_t^\top)/(2t^2)}. \tag{35}$$

Chose $\epsilon = 1/2$ and define

$$\boldsymbol{G}_t := \mathbb{E}\phi(\boldsymbol{X}\boldsymbol{w})\phi(\boldsymbol{X}\boldsymbol{w})^\top 1\{\|\phi(\boldsymbol{X}\boldsymbol{w})\| \le t\}. \tag{36}$$

The inequality becomes

$$\mathbb{P}\left\{\lambda_{\min}(\boldsymbol{B}_t\boldsymbol{B}_t^\top) \le m\lambda_{\min}(\boldsymbol{G}_t)/2\right\} \le k e^{-m\lambda_{\min}(\boldsymbol{G}_t)/(8t^2)}. \tag{37}$$

If we choose $m \ge \frac{8t^2}{\lambda_{\min}(\boldsymbol{G}_t)}\log(\frac{k}{\delta})$, then with probability at least $1 - \delta$, we have

$$\lambda_{\min}(\boldsymbol{B}_t\boldsymbol{B}_t^\top) \ge m\lambda_{\min}(\boldsymbol{G}_t)/2 \tag{38}$$

Note that

$$\|\boldsymbol{G}^\infty - \boldsymbol{G}_t\| = \|\mathbb{E}\left[\phi(\boldsymbol{X}\boldsymbol{w})\phi(\boldsymbol{X}\boldsymbol{w})^T\right] - \phi(\boldsymbol{X}\boldsymbol{w})\phi(\boldsymbol{X}\boldsymbol{w})^T 1\{\|\phi(\boldsymbol{X}\boldsymbol{w})\| \le t\}]\| \tag{39}$$

$$\le \mathbb{E}\|\left[\phi(\boldsymbol{X}\boldsymbol{w})\phi(\boldsymbol{X}\boldsymbol{w})^T\right] - \phi(\boldsymbol{X}\boldsymbol{w})\phi(\boldsymbol{X}\boldsymbol{w})^T 1\{\|\phi(\boldsymbol{X}\boldsymbol{w})\| \le t\}]\|, \quad (i) \tag{40}$$

$$= \mathbb{E}\left[\|\phi(\boldsymbol{X}\boldsymbol{w})\|^2 \cdot 1\{\|\phi(\boldsymbol{X}\boldsymbol{w})\| > t\}\right] \tag{41}$$

$$\le \int_0^\infty \mathbb{P}\left(\|\phi(\boldsymbol{X}\boldsymbol{w})\|^2 \cdot 1\{\|\phi(\boldsymbol{X}\boldsymbol{w})\| > t\} \ge s\right) ds \tag{42}$$

Since $\phi$ is Lipschitz continuous, $\|\phi(\boldsymbol{X}\boldsymbol{w})\|$ is a sub-Gaussian random variable with coefficient $C\|\boldsymbol{X}\|$, where the constant $C > 0$ only depends on the Lipschitz coefficient of $\phi$. Then we obtain

$$\|\boldsymbol{G}^\infty - \boldsymbol{G}_t\| \le \int_0^\infty e^{-c_0(s+t^2)/(2\|\boldsymbol{X}\|^2)}ds \le e^{-c_0 t^2/(2\|\boldsymbol{X}\|^2)} \cdot \frac{2\|\boldsymbol{X}\|^2}{c_0} \tag{43}$$

where $c_0 > 0$ is some absolute constant.

We can choose $t > 0$ small so that the RHS is less than $\lambda_0/2$. Specifically, we choose

$$t = \sqrt{\frac{2\|\boldsymbol{X}\|^2}{c_0}\log\left(\frac{4\|\boldsymbol{X}\|^2}{c_0\lambda_0}\right)} \tag{44}$$

589 and obtain

$$\|\boldsymbol{G}^\infty - \boldsymbol{G}_t\| \leq \lambda_0/2. \tag{45}$$

590 This yields

$$\lambda_{\min}(\boldsymbol{G}_t) \geq \lambda_{\min}(\boldsymbol{G}^\infty) - \|\boldsymbol{G}^\infty - \boldsymbol{G}_t\| \geq \lambda_0/2. \tag{46}$$

591 Altogether, we have

$$\lambda_{\min}(\boldsymbol{A}\boldsymbol{A}^\top) \geq \lambda_{\min}(\boldsymbol{B}_t\boldsymbol{B}_t) \geq m\lambda_{\min}(\boldsymbol{G}_t) \geq m\lambda_0/4. \tag{47}$$

592 **Lemma 6.** *Suppose $\phi$ is Lipschitz continuous and $\lambda_0 > 0$. Then, for any $\delta > 0$, if $m =$*
593 $\Omega(\frac{k}{\lambda_0}\log(\frac{k}{\lambda_0})\log(\frac{k}{\delta}))$, *then with probability at least $1 - \delta$, we have $\lambda_{\min}(\boldsymbol{G}) \geq \lambda_0/4$. Hence,*
594 *the feature matrix $\boldsymbol{H}$ has linearly independent rows.*

595 Therefore, combining Lemma 5 with Lemma 6 yields the desired result in Proposition 2.

### C.3 Expressive GNNs with Injective Message Passing

597 Recall the foundational result from [50], as restated below.

598 **Theorem 2.** *[50, Lemma 2 and Theorem 3] Suppose the aggregation and combination functions are*
599 *injective. Then a GNN returns different embeddings for two given graphs if and only if the 1-WL test*
600 *decides non-isomorphic.*

601 Combining Propostion 1, Propostion 2, and Theorem 2 yields Theorem 1.

## D   Missing Proofs for Section 5

### D.1   Depth Analysis for Message Passing Layers

604 Let us first consider the fully connected graph $K_n$. We will assume each node initially has the
605 same node features or colors. Since all nodes are connected, each $u$ receives an identical multiset
606 of neighbor colors. Combining with its unique color, the resulting color remains the same for all
607 $u$. No further refinement occurs because the colors are stabilized. This implies that GNN achieves
608 the maximal expressive power using one message passing scheme, since it simulates the 1-WL test
609 iteration using an injective message passing operation.

610 **Lemma 7.** *For the complete graph $K_n$ with identical initial node colors, the 1-WL test stabilizes after*
611 *one iteration. Consequently, a GNN with one message-passing layer achieves maximal expressive*
612 *power on $K_n$.*

613 Now, let us consider the path graph $P_n$ with nodes $\{v_1, \cdots, v_n\}$ and edges $(v_i, v_{i+1})$. Assume
614 all nodes start with their identical initial colors, *i.e.*, $c_u^0 = c$ for all $u \in V$. At the first iteration,
615 endpoints $v_1$ and $v_n$ (degree 1) receive multiset $\{c^{(0)}\}$, while the internal nodes $\{v_2, \cdots, v_{n-1}\}$
616 (degree 2) receive multiset $\{c^{(0)}, c^{(0)}\}$. Hence, endpoints and internal nodes are assigned distinct
617 colors after hashing: the endpoints have new colors while the internal nodes retain $c$, not refined yet.
618 At iteration $k$, nodes within $k$ hops of an endpoint have refined their colors. The "new endpoints" $v_k$
619 and $v_{n-k+1}$ are assigned new colors, while the rest internal nodes remain unchanged. Hence, the
620 color propagates inward until the midpoints stabilize. Hence, we need $\lceil n/2 \rceil$ iterations to stabilize
621 all colors. Consequently, GNNs with injective message passing need $\lceil n/2 \rceil$ message passes to
622 distinguish all nodes in $P_n$, achieving the maximal expressive power.

623 **Lemma 8.** *For the path graph $P_n$ with identical initial colors, the 1-WL test stabilizes after $\lceil n/2 \rceil$*
624 *iterations. Consequently, a GNN needs $\lceil n/2 \rceil$ message-passing layers to achieve maximal expressive*
625 *power on $P_n$.*

626 Consider a complete binary tree with $n = 2^h - 1$ nodes, where $h$ is the height of the tree. When all
627 nodes begin with identical initial colors, the 1-WL test exhibits a characteristic bottom-up refinement
628 process. At the first iteration, the leaves (degree 1) become distinguishable from internal nodes
629 (degree 2 or 3 for the root), receiving new colors. In contrast, internal nodes retain their original color
630 (except probably the root). This creates the first level of differentiation at the tree's lowest level.

The refinement proceeds upward through subsequent iterations. At each step $k$, nodes at height $k$ from the bottom become distinguishable because they now see distinct color patterns in their subtrees. Specifically, parents of already-distinguished child nodes receive new colors based on their children's unique color configurations, while higher-level nodes remain unchanged until their turn in this propagation process.

At the second iteration, the parents-of-leaves nodes now see their children have a special leaf color. Hence, these parents get a new color while the rest internal nodes remain unchanged. Remarkably, it essentially treats the parents-of-leaves nodes as "new leaves" and assigns new "leaf" colors to them. Hence, each iteration reveals one more level of the hierarchy, and the refinement proceeds upward from leaves to root. Hence, the total number of iterations equals the tree height $h = \Theta(\log n)$.

**Lemma 9.** *For a balanced binary tree graph with $n$ nodes with identical initial colors, the 1-WL test stabilizes after $\Theta(\log n)$ iterations. Consequently, a GNN needs $\Theta(\log n)$ message-passing layers to achieve maximal expressive power.*

Consider a $\sqrt{n} \times \sqrt{n}$ grid graph, where all nodes start with identical initial colors, *i.e.*, $c_u^{(0)} = c$ for all $u \in V$. At the first iteration, the corner nodes (degree 2) receive a multiset $\{c^{(0)}, c^{(0)}\}$ and the boundary nodes (degree 3) receive a multiset $\{c^{(0)}, c^{(0)}, c^{(0)}\}$ from their neighbors, while the rest internal nodes receive $\{c^{(0)}, c^{(0)}, c^{(0)}, c^{(0)}\}$. Hence, we can assign two new colors to the corner and boundary nodes, while the color of the internal nodes remains unchanged. At the $k$-th iteration, nodes at distance $k-1$ from the boundary are refined. The new color refinement propagates inward from the boundary at a rate of one layer per iteration. Hence, the process stabilizes when it reaches the center of the grid. As a result, we need $\lceil \sqrt{n}/2 \rceil$ iterations to stabilize the colors, since the maximum distance from the boundary to the center is $\lceil \sqrt{n}/2 \rceil$.

**Lemma 10.** *For a $\sqrt{n} \times \sqrt{n}$ gride graph with identical initial colors, the 1-WL test stabilizes after $\Theta(\sqrt{n})$ iterations. Consequently, a GNN needs $\Theta(\log n)$ message-passing layers to achieve maximal expressive power.*

## D.2 Stability of Deep Message Passing

In this section, we prove the iterative relation of $\mathbb{E}[\|\boldsymbol{H}^{(\ell)}\|^2]$ in Lemma 1. Recall that the proposed ML has the following matrix form:

$$\boldsymbol{H}^{(\ell)} = ((1+\epsilon)\boldsymbol{I}_n + \boldsymbol{A}) \frac{1}{\sqrt{m}} \phi(\boldsymbol{H}^{(\ell-1)}\boldsymbol{W}^{(\ell)}). \tag{48}$$

As each layer uses independent $\boldsymbol{W}^{\ell}$, we can first assume $\boldsymbol{H}^{(\ell-1)}$ is fixed. Then we have

$$\mathbb{E}\left[\|\boldsymbol{H}^{(\ell)}\|_F^2 \mid \boldsymbol{H}^{(\ell-1)}\right]$$
$$=\frac{1}{m}\operatorname{Tr}\left(\boldsymbol{T}^2 \mathbb{E}\phi(\boldsymbol{H}^{(\ell-1)}\boldsymbol{W}^{(\ell)})\phi(\boldsymbol{H}^{(\ell-1)}\boldsymbol{W}^{(\ell)})^\top\right)$$
$$=\frac{1}{m}\sum_{r=1}^m \operatorname{Tr}\left(\boldsymbol{T}^2 \mathbb{E}\phi(\boldsymbol{H}^{(\ell-1)}\boldsymbol{w}_r)\phi(\boldsymbol{H}^{(\ell-1)}\boldsymbol{w}_r)^\top\right)$$
$$\leq\frac{K^2}{m}\sum_{r=1}^m \operatorname{Tr}\left(\boldsymbol{T}^2 \boldsymbol{H}^{(\ell-1)}\mathbb{E}\left[\boldsymbol{w}_r\boldsymbol{w}_r^\top\right]\boldsymbol{H}^{(\ell-1)\top}\right)$$
$$=\sigma_\ell^2 K^2 \operatorname{Tr}\left(\boldsymbol{T}^2 \boldsymbol{H}^{(\ell-1)}\boldsymbol{H}^{(\ell-1)\top}\right)$$
$$\leq\sigma_\ell^2 K^2 \|\boldsymbol{T}\|^2 \|\boldsymbol{H}^{(\ell-1)}\|_F^2,$$

where $\boldsymbol{T} = (1+\epsilon)\boldsymbol{I}_n + \boldsymbol{A}$ and we use the Lipschitz continuity of $\phi$. Adding the expectation on both sides for $\boldsymbol{H}^{\ell-1}$ yields the desired result.

## E Experiments Setup and Additional Results

Since the supplemental submission deadline is one week after the full paper deadline, we only include the experimental setups here. Please refer to the additional results in the supplemental files.

Following [33], we select eight node classification datasets to evaluate our method. Cora, CiteSeer, and PubMed are widely used benchmark datasets for evaluating citation networks [38]. Following the semi-supervised learning setup described in [28], we apply the same data splits and evaluation metrics. Additionally, we include the Computer and Photo datasets from [39], which represent co-purchase networks where nodes correspond to products and edges indicate frequent co-purchases. We also consider the CS and Physics datasets from [39], which are co-authorship networks where nodes represent authors, and edges signify collaborative publications. For these datasets, we adopt the standard 60%/20%/20% split for training, validation, and testing, using accuracy as the evaluation metric [6, 40, 8, 33]. Lastly, we evaluate on the WikiCS dataset, leveraging its official data splits and metrics as specified in [34]. We perform hyperparameter tuning for all experiments, following the search space defined in [8, 33]. Specifically, we employ the Adam optimizer [27] with learning rates selected from 0.001, 0.005, 0.01 and a maximum of 2500 training epochs. The hidden dimension is tuned over 64, 256, 512, while dropout rates are chosen from 0.2, 0.3, 0.5, 0.7. We also explore the number of message-passing layers and feature-learning layers within the range of 1, 2, 3, 4, 5, 6, 7, 8, 9, 10. All reported results represent the mean and standard deviation across five independent runs with different random initializations. For baseline comparisons, we utilize the official code provided by [33]. All experiments are tested on a server with a CPU AMD Threadripper 2990WX, a GPU Nvidia RTX 4090, and 128GB of memory.

