# OpenReview forum: "Does Depth Really Hurt GNNs? Injective Message Passing Enables Deep Graph Learning"
_NeurIPS.cc/2025/Conference — Submitted to NeurIPS 2025_

### Official Review · Reviewer_hqYh · 2025-07-02

**Clarity:** 3
**Significance:** 4
**Originality:** 3
**Rating:** 5
**Confidence:** 3

**Summary:**

This paper investigates the problem of oversmoothing in deep GNNs. The authors demonstrate that oversmoothing is a problem of non-injective message passing in classic message-passing GNN architecture. They address this problem by proposing a provably injective message passing layer and further prove that it matches the expressive power of the WL test by design. Based on this, they propose a decoupled GNN architecture and provide a theoretical analysis of its expressivity and stability. Evaluations on several node classification tasks demonstrate a favorable performance compared to several classical GNN architectures.

**Questions:**

I could not follow all parts of the paper in detail. In particular, it could be helpful to explain how the decoupled architecture can be mapped to the functions in classical GNNs.


[Q1] In Section 4.2., A is introduced as the “lifted feature matrix”. However, A is also introduced as the adjacency matrix in Section 3. Are both the same? I.e., message passing is changed from following the adjacency matrix to following the lifted feature matrix?

[Q2] How does the proposed decoupled architecture fit into the GNN model in Section 3? Does the message passing layer affect the AGGREGATE function, the COMBINE function, or both?

**Ethical Concerns:**

["NO or VERY MINOR ethics concerns only"]

**Final Justification:**

Most of my questions could be resolved in the discussion. There are points, especially in terms of the experiments, that could be improved, but overall, this is not a critical issue, as the evaluations are still solid.

**Limitations:**

yes

**Quality:**

3

**Strengths And Weaknesses:**

Strengths

[S1] Thorough analysis of an important problem. While the oversmoothing problem is observed in the literature, it is still not known why it occurs. This paper makes a significant step to understand the problem in more depth.

[S2] The proposed decoupled GNN architecture is, to the best of my knowledge, new. It is theoretically and empirically analyzed.

[S3] The paper is well-written and makes a good job explaining both the problem and the solution, providing both a good intuition and a thorough description of the proposed architecture.


Opportunities for improvement:

[O1] While the method is empirically evaluated on some datasets, it is confined to node classification tasks and rather small graphs. A higher variety of tasks could provide further insights. Also, empirical evaluations do not report the training time and memory consumption, which is an important consideration in real-world applications.

[O2] Presentation: Figures 1 and 2 are too dense and have too thin lines and too small fonts. It is hard to recognize anything in them even when zooming in.

---

> ### Author Rebuttal · Authors · 2025-07-27
>
> We thank the reviewer for the constructive and thoughtful feedback. Below we provide our responses to the main points raised.
>
> **[O1] Limited task coverage and efficiency evaluation**: We appreciate the reviewer’s suggestion to broaden the empirical scope. While our main focus was on node classification to isolate the effects of injectivity and expressivity, we agree that expanding the empirical scope to include additional tasks and datasets offers valuable insight.
>
> To address this, we conducted additional experiments during the rebuttal period:
> **1. Graph Classification:** We evaluated our method on five standard benchmarks (MUTAG, PTC, NCI1, PROTEINS, and REDDIT-B). As shown below, our method achieves competitive or superior performance across diverse domains, reinforcing its broad applicability. Due to space and time constraints, we have not yet evaluated link prediction or heterophilous graphs, but we recognize their importance and will include them in the revised paper.
>
> > We use the same tuning strategy as for node classification in the paper. Bolded values indicate the best performance per dataset.
>
> | Model        | MUTAG           | PTC             | NCI1            | PROTEINS         | REDDIT-B         |
> |--------------|------------------|------------------|------------------|------------------|------------------|
> | # Graphs     | 188              | 344              | 4110             | 1113             | 1000             |
> | # Classes    | 2                | 2                | 2                | 2                | 2                |
> | Avg. # Nodes | 17.9             | 25.5             | 29.8             | 39.1             | 19.8             |
> | **GCN**      | 0.9257 ± 0.06    | 0.7850 ± 0.04    | **0.8474 ± 0.01**| 0.7943 ± 0.02    | 0.9130 ± 0.02    |
> | **SAGE**     | 0.9099 ± 0.07    | 0.7707 ± 0.06    | 0.8409 ± 0.01    | 0.7898 ± 0.03    | 0.7625 ± 0.02    |
> | **GAT**      | 0.9260 ± 0.07    | 0.7939 ± 0.04    | 0.8345 ± 0.01    | 0.7853 ± 0.02    | 0.7896 ± 0.03    |
> | **GIN**      | 0.9523 ± 0.06    | 0.7641 ± 0.05    | 0.8431 ± 0.01    | 0.7763 ± 0.02    | 0.9190 ± 0.01    |
> | **GraphConv**| 0.9471 ± 0.06    | 0.7618 ± 0.06    | 0.8494 ± 0.02    | 0.7826 ± 0.03    | 0.9015 ± 0.01    |
> | **Ours**     | **0.9576 ± 0.06**| **0.8082 ± 0.02**| 0.8380 ± 0.01    | **0.7997 ± 0.02**| **0.9265 ± 0.02**|
>
> **2. Scalability and Oversmoothing Robustness**
> We further tested our model on node classification tasks across three graph sizes—Cora (small), Pubmed (medium), and OGBN-Arxiv **(large)**—under oversmoothing mitigation setups (residual connections and normalization). Our method consistently performs strongly, even without architectural tweaks, confirming its robustness across graph regimes.
>
> | Model         | Cora (2.7K)      | Pubmed (19.7K)   | OGBN-Arxiv (169K) |
> | ------------- | ---------------- | ---------------- | ----------------- |
> | **GCN**       | 83.16 ± 1.14     | 78.90 ± 0.93     | 69.92 ± 0.01      |
> | + Norm        | 80.64 ± 0.26     | 77.32 ± 0.66     | **72.74 ± 0.05**  |
> | + Res         | 82.48 ± 1.64     | 77.50 ± 0.83     | 71.80 ± 0.06      |
> | **SAGE**      | 80.56 ± 1.34     | 77.10 ± 0.60     | 70.78 ± 0.02      |
> | + Norm        | 76.12 ± 2.61     | 74.26 ± 1.12     | 72.50 ± 0.07      |
> | + Res         | 80.68 ± 0.73     | 76.84 ± 1.69     | 71.04 ± 0.06      |
> | **GAT**       | 81.92 ± 1.33     | 77.02 ± 1.13     | 69.93 ± 0.06      |
> | + Norm        | 78.70 ± 0.23     | 76.30 ± 1.30     | 72.15 ± 0.02      |
> | + Res         | 83.06 ± 0.77     | 76.38 ± 0.83     | 71.10 ± 0.13      |
> | **GIN**       | 79.96 ± 1.42     | 75.14 ± 0.90     | 36.32 ± 0.38      |
> | + Norm        | 69.82 ± 3.62     | 74.82 ± 1.26     | 70.86 ± 0.26      |
> | + Res         | 80.30 ± 0.97     | 72.88 ± 3.12     | 47.13 ± 2.55      |
> | **GraphConv** | 82.08 ± 1.09     | 76.40 ± 0.76     | 69.52 ± 0.15      |
> | + Norm        | 76.40 ± 1.45     | 74.06 ± 2.02     | 72.60 ± 0.25      |
> | + Res         | 81.58 ± 0.31     | 75.96 ± 1.20     | 70.96 ± 0.13      |
> | **Ours**      | **83.74 ± 0.55** | **79.16 ± 1.23** | 72.71 ± 0.02      |
> | + Norm        | 75.28 ± 2.24     | 76.81 ± 0.83     | **72.73 ± 0.06**  |
> | + Res         | 83.38 ± 1.53     | **79.56 ± 1.28** | 72.72 ± 0.38      |
> > To prevent overfitting, all GNNs are tuned with dropout. Bolded values indicate the best performance per dataset.
>
> **3. Efficiency (Runtime and Memory):** Our architecture uses standard linear message passing followed by a one-layer MLP, resulting in time and memory complexity comparable to GCN—scaling linearly with the number of edges and hidden dimensions. To support this, we benchmarked training time and GPU memory usage on the Cora dataset across all evaluated GNNs. As shown below, our method is as efficient as lightweight models like GCN and SAGE, and significantly more resource-friendly than attention-based models like GAT. These results reinforce that our approach is not only expressive and stable, but also practical and scalable for real-world deployment.
>
> **Training time per epoch (Cora, in seconds)**
> | Model      | GCN    | SAGE   | GAT    | GIN    | GraphConv | Ours       |
> | ---------- | ------ | ------ | ------ | ------ | --------- | ---------- |
> | Time/epoch | 0.1100 | 0.0902 | 0.1532 | 0.1074 | 0.1000    | **0.0933** |
>
> **GPU Peak Memory Usage (Cora, in GB)**
> | Model       | GCN  | SAGE | GAT  | GIN  | GraphConv | Ours     |
> | ----------- | ---- | ---- | ---- | ---- | --------- | -------- |
> | Memory (GB) | 0.05 | 0.04 | 0.28 | 0.05 | 0.04      | **0.04** |
>
> **[O2] Presentation quality of Figures 1 and 2**: Thank you for the helpful feedback. We will revise Figures 1 and 2 to use larger fonts, thicker lines, and clearer layout to improve readability in the final version.
>
> **[Q1] Clarification on the meaning of matrix A**: We thank the reviewer for noticing this ambiguity. The two instances of $A$ refer to different objects: one is the adjacency matrix (used in message passing), and the other is the lifted feature matrix (used to define the injective map). We apologize for the confusion and will revise the notation in the final version to clearly distinguish them.
>
> **[Q2] Role of decoupled architecture in the GNN**: Thank you for the question. Our injective message passing layer (Eq. 7–8) differs from classical GNN layers by introducing an MLP-based feature lifting step **before** aggregation. Specifically, we first apply a one-layer MLP to map input features into a linearly independent space, followed by a linear aggregation where both AGGREGATE and COMBINE steps are additive (i.e., sum-based). This design ensures injectivity (with high probability), as supported by our theoretical analysis and numerical experiments.
>
> In the broader decoupled architecture, we separate message passing (to encode structure) from feature learning (to extract task-specific representations). After a shallow stack of injective MP layers, we stop message passing and use pure MLP layers for deeper transformations. This avoids oversmoothing in deep MP stacks while preserving the benefit of depth in feature learning—motivated by our observation that structural information saturates early but deeper MLPs may still improve classification.

---

> > ### Comment · Reviewer_hqYh · 2025-08-04
> >
> > Thank you for the additional results and clarifications.
> >
> > Regarding the graph datasets, OGBN considers the arxiv dataset still as "small" scale (https://ogb.stanford.edu/docs/nodeprop/). Large-scale would be papers100M which is 1,000 times larger. So in that respect, the argumentation / result is not fully convincing.
> >
> > Also, the system measurements (time and memory) are performed on a tiny graph (cora) which runs through in less than a second. To get representative results, this should be repeated with larger graphs where time and memory really matter.
> >
> > Thanks for the clarifications, which I appreciate.

---

> > > ### Author Response · Authors · 2025-08-04
> > >
> > > Thank you for the helpful feedback. We appreciate the clarification on scale and agree that larger datasets like papers100M offer stronger insights. Due to rebuttal constraints, we couldn’t include further results, though OGBN-Arxiv remains a meaningful step beyond small graphs. We plan to extend to larger benchmarks in the revised version.

---

### Official Review · Reviewer_1KdM · 2025-07-03

**Clarity:** 4
**Significance:** 2
**Originality:** 2
**Rating:** 2
**Confidence:** 5

**Summary:**

The authors propose a GNN architecture, which shows Weisfeiler-Leman expressivity with a random initialization of parameters. They investigate freezing the random parameters, just training the last layers for a prediction task, and obtain competitive results in experiments. The architecture is not to be susceptible to oversmoothing.

**Questions:**

**Q1)** Could you elaborate why the proposed architecture has desirable properties that are not exhibited similarly by existing architectures by Morris et al. (linear transformation) and GIN?

**Ethical Concerns:**

["NO or VERY MINOR ethics concerns only"]

**Final Justification:**

Due to the claims made by the authors in their rebuttal, I have checked the literature on set functions, specifically [1], where the consequences for standard MPNNs are even made explicit. This limits the novelty of the approach further. Moreover, the introduction and the motivation of the proposed architecture, as well as the interpretation of the experimental results, need to be significantly revised. In the rebuttal, the authors argue that their approach is more general and allows application to further MPNN architectures different from those in [1]. However, this is not part of the submitted paper. Overall, in my opinion, the paper is not ready for publication in its current form but has great potential after a major revision.

Therefore, I kept my overall rating. As a consequence of my literature research, I have increased my confidence score.

[1] T. Amir et al., Neural Injective Functions for Multisets, Measures and Graphs via a Finite Witness Theorem, NeurIPS 2023.

**Limitations:**

yes

**Quality:**

2

**Strengths And Weaknesses:**

## Strengths
**S1)** The paper is exceptionally well written and a pleasure to read. All aspects are clearly expressed and comprehensible.

**S2)** Investigating the contradiction between theoretical WL expressivity and the phenomenon of oversmoothing is interesting and deserves further investigation.

**S3)** Proposition 2 provides a valuable bound on the probability that a linear layer yields linearly independent node features.

## Weaknesses
**W1)** The paper is of limited novelty, and besides Proposition 2, it is hard to find original contributions except for the excellent write-up:
 * The proposed architecture appears to be a standard GNN architecture, simply using a linear layer to transform embeddings. Although presented a bit differently, I do not see any fundamental difference between widely-used approaches suggested, e.g., in Christopher Morris, Martin Ritzert, Matthias Fey, William L. Hamilton, Jan Eric Lenssen, Gaurav Rattan, Martin Grohe: Weisfeiler and Leman Go Neural: Higher-Order Graph Neural Networks. AAAI 2019: 4602-4609
 * It is already known that a linear layer with random initialization leads to injective set functions with high probability (see Tal Amir, Steven J. Gortler, Ilai Avni, Ravina Ravina, Nadav Dym: Neural Injective Functions for Multisets, Measures and Graphs via a Finite Witness Theorem. NeurIPS 2023). It would be good to put the results into this context. Moreover, this result implies that GIN with random parameters achieves 1-WL expressivity with high probability. This limits the novelty of the approach and makes it advisable to include GIN in the experiments as a baseline.

**W2)** The idea of summing linearly independent vectors to obtain injective multiset functions is not put in the context of the large body of work on set functions. In particular, a discussion of the required embedding dimension would strengthen the paper.

**W3)** The experiments could be extended to strengthen the paper. It would be good to include GIN in Table 1 (with and without frozen parameters, see above). In the experiments, the authors also chose not to freeze the random parameters used in the aggregation step and claim that this is not problematic because of their novel architecture. I cannot follow this argument since I do see a fundamental difference from existing architectures. More experiments showing that GIN and the linear layer-based architecture by Morris et al. suffer from oversmoothing compared to the new architecture are required.

## Minor remarks
 * l125: The given references do not assume identical node features.
 * The paper states that the proposed architecture is "injective by design" or "provably injective" -- as this only holds with high probability for certain hyperparameter choices, the formulation is not adequate.
 * Figure 2 is not readable due to the font size.
 * The approach suggests that embeddings encoding WL colors, followed by an MLP, have desirable properties. However, it is not clear why the embedding step should be performed by a GNN-like architecture (with frozen parameters) at all instead of some algorithm such as color refinement. It would be good to make the advantages explicit.

---

> ### Author Rebuttal · Authors · 2025-07-25
>
> **W1) Limited Novelty (Proposition 2 is similar to [1,2])**: We appreciate the reviewer pointing out this connection. After studying [1-2], we refined and strengthened our theoretical results by incorporating their insights.
>
> Specifically, Corollary 8 in [1] shows that injectivity in the aggregation step alone is sufficient to achieve WL-level expressivity, and the combination function is not essential. This observation allows us to generalize our result to a **broader** class of GNN architectures. In our setting, Proposition 1 shows that linear aggregation, including the use of self-loops, is injective if the unique inputs are linearly independent. Normalization (as in GCN, GIN, GraphSAGE) and MLPs following aggregation can then be viewed as part of a *nonlinear lifting function*. As long as the normalization **preserve pairwise distinctness**, our Proposition 2 guarantees that the lifted features are linearly independent with high probability, provided the width is $\tilde{\Omega}(k)$.
>
> This generalization implies that a wide range of GNNs using summation-based aggregation followed by nonlinear transformations are likely to be injective. It offers a unified theoretical explanation for why these models can achieve injectivity without requiring complex architectural changes. It also helps explain why many expressivity-focused GNNs (e.g., higher-order variants) often show limited empirical gains: in practice, standard GNNs with sufficient width may already match 1-WL expressivity on real-world graphs. Our result thus highlights both the theoretical and practical boundaries of improving expressivity.
>
> **W2) Lack of comparison with [1,2] and discussion of embedding dimension**: We agree that our original submission lacked sufficient discussion of [1,2] and embedding dimension, and we will include a more detailed comparison in the revised paper. Briefly:
> - [1] provides an *existence* of a weight matrix $W$ that ensures injectivity of summation aggregation result (Lemma 9-11), and establishes WL-expressivity (Theorem 2 and 6). However, they do not show whether standard random initialization achieves this in practice. In contrast, our Proposition 2 proves that **common random initializations (e.g., He/Xavier)** yield injectivity with high probability, providing a constructive and practical guarantee.
>
> - [2] does analyze injectivity under random features, but with stricter conditions: requiring a width to be as large as $2n(d + 1) + 1$, where $n$ is the number of nodes and $d$ is the input dimension. Our approach only requires the width to be **near-linear** in $k$, the number of distinct input embeddings. When stacking layers, $k$ corresponds to the number of *distinct rooted subtrees*, which often remains small relative to $n$ and $d$, yielding better scalability on large graphs.
>
> Moreover, as discussed under *W1*, with a small tweak in assumptions, our framework can be extended to many existing GNNs, as long as they (i) use summation-based aggregation, and (ii) preserve pairwise distinctness post-normalization. This significantly broadens the applicability and novelty of our result.
>
> **W3)** *Lack of comparison with GIN [6] and GraphConv [2] on Oversmoothing*: We thank the reviewer for highlighting this concern. To directly address it, we first conducted *100-layer initialization experiments* comparing GIN/GraphConv/GCN/GAT/SAGE, and our method, all *without skip connections, normalization, or dropout*, to isolate message passing effects. Our findings show:
>
> - **GCN, GAT, and SAGE** suffer from **oversmoothing**, with Dirichlet energy and embedding norm decay, and embedding rank collapses to zero by ~50 layers.
> - **GIN and GraphConv** exhibit severe **numerical instability**, with exploding energy and norms, yielding `inf/NaN` values as early as 30 layers.
> - **Our method** maintains stable energy, norm, and full rank beyond 50 layers, degrading gracefully even at depth 100.
>
> Due to rebuttal response constraints, we report  Dirichlet energy and embedding rank on the Cora dataset as robust metrics here.  Full results on other datasets will be included in the revised paper. These results clarify that oversmoothing and instability are common failure modes in deep GNNs, and our method is significantly more stable at large depths.
>
> | Layer | Metric  | GCN    | GAT    | GIN   | SAGE   | GraphConv | Ours   |
> | ----- | ------ | ------ | ------ | ----- | ------ | --------- | ------ |
> | 001   | log(E) | 3.48   | 5.15   | 7.21  | 4.67   | 9.61      | 5.96   |
> |       | Rank   | 100    | 36     | 37    | 33     | 37        | 37     |
> | 020   | log(E) | -28.93 | -20.81 | 38.72 | -29.33 | 73.84     | -7.17  |
> |       | Rank   | 100    | 100    | 100   | 100    | 100       | 100    |
> | 050   | log(E) | -72.60 | -62.29 | inf   | -83.61 | inf       | -29.31 |
> |       | Rank   | 1      | 1      | NaN   | NaN      | NaN       | 100    |
> | 100   | log(E) | -inf   | -inf   | inf   | -inf   | inf       | -63.46 |
> |       | Rank   | NaN      | NaN      | NaN   | NaN      | NaN       | 1      |
> *(log(E) = log Dirichlet energy; Rank = embedding rank)*
>
> Beyond initialization, we also included GIN and GraphConv in our node classification experiments. This is consistent with prior literature noting their vulnerability to depth and instability.
>
> Beyond initialization, we included GIN and GraphConv in our node classification and oversmoothing mitigation experiments on Cora, Pubmed, and OGBN-Arxiv. Both models consistently underperformed compared to other GNNs, including ours—consistent with prior findings on their sensitivity to depth and training instability. While residual connections and normalization modestly improved baseline models, our method achieved strong performance without relying on such architectural fixes, highlighting its inherent stability and scalability across diverse graph regimes.
>
> | Model         | Cora (2.7K)      | Pubmed (19.7K)   | OGBN-Arxiv (169K) |
> | ------------- | ---------------- | ---------------- | ----------------- |
> | **GCN**       | 83.16 ± 1.14     | 78.90 ± 0.93     | 69.92 ± 0.01      |
> | + Norm        | 80.64 ± 0.26     | 77.32 ± 0.66     | **72.74 ± 0.05**  |
> | + Res         | 82.48 ± 1.64     | 77.50 ± 0.83     | 71.80 ± 0.06      |
> | **SAGE**      | 80.56 ± 1.34     | 77.10 ± 0.60     | 70.78 ± 0.02      |
> | + Norm        | 76.12 ± 2.61     | 74.26 ± 1.12     | 72.50 ± 0.07      |
> | + Res         | 80.68 ± 0.73     | 76.84 ± 1.69     | 71.04 ± 0.06      |
> | **GAT**       | 81.92 ± 1.33     | 77.02 ± 1.13     | 69.93 ± 0.06      |
> | + Norm        | 78.70 ± 0.23     | 76.30 ± 1.30     | 72.15 ± 0.02      |
> | + Res         | 83.06 ± 0.77     | 76.38 ± 0.83     | 71.10 ± 0.13      |
> | **GIN**       | 79.96 ± 1.42     | 75.14 ± 0.90     | 36.32 ± 0.38      |
> | + Norm        | 69.82 ± 3.62     | 74.82 ± 1.26     | 70.86 ± 0.26      |
> | + Res         | 80.30 ± 0.97     | 72.88 ± 3.12     | 47.13 ± 2.55      |
> | **GraphConv** | 82.08 ± 1.09     | 76.40 ± 0.76     | 69.52 ± 0.15      |
> | + Norm        | 76.40 ± 1.45     | 74.06 ± 2.02     | 72.60 ± 0.25      |
> | + Res         | 81.58 ± 0.31     | 75.96 ± 1.20     | 70.96 ± 0.13      |
> | **Ours**      | **83.74 ± 0.55** | **79.16 ± 1.23** | 72.71 ± 0.02      |
> | + Norm        | 75.28 ± 2.24     | 76.81 ± 0.83     | **72.73 ± 0.06**  |
> | + Res         | 83.38 ± 1.53     | **79.56 ± 1.28** | 72.72 ± 0.38      |
> > To prevent overfitting, all GNNs are tuned with dropout. Bolded values indicate the best performance per dataset.
>
> Together, these results reinforce our empirical claims and clarify the practical advantages of our method over existing architectures. We will include full results in the camera-ready version if accepted.
>
> **Minor 1)**:  Thank you for pointing this out. It was a typo and we meant that the WL test and GNNs use the same node features as their input coloring. We will revise the sentence to clarify this.
>
> **Minor 2)**: We thank the reviewer for the helpful clarification. To avoid confusion, we will revise our statements to be more precise, e.g., stating that randomly initialized message passing layers are injective with high probability, provided the embedding dimension is sufficiently large.
>
> **Minor 3)**: Thank you for the feedback. We will increase the font size in Figure 2 to improve readability in the revised version.
>
> **Minor 4)**: We appreciate this insightful remark. Indeed, as in the WL graph kernel [3], color refinement alone can offer good embedding and performances without GNNs. While integrating WL colorings with node features in downstream models is theoretically plausible, this has not been widely explored in practice. Our use of a GNN encoder aligns with common practice in graph learning, where message-passing layers encode local structure and adapt to task signals. As shown in Figures 1c and 2b, freezing message passing introduces noise due to random initialization, motivating the need for training. As future work, we plan to extend our theoretical framework beyond initialization by building on recent insights from NTK theory [4] and feature learning results for pure MLPs [5].
>
> [1] C. Morris et al., Weisfeiler and Leman Go Neural: Higher-order Graph Neural Networks, AAAI 2019.
>
> [2] T. Amir et al., Neural Injective Functions for Multisets, Measures and Graphs via a Finite Witness Theorem, NeurIPS 2023.
>
> [3] Shervashidze, Nino, et al., Weisfeiler-lehman graph kernels, JMLR 2011
>
> [4] Jacot, Arthur, et al., Neural tangent kernel: Convergence and generalization in neural networks, NeurIPS 2018
>
> [5] Chen, Zixiang, et al., Global Convergence and Rich Feature Learning in $L$-Layer Infinite-Width Neural Networks under $\mu$ P Parametrization, ICML 2025
>
> [6] Xu, Keyulu, et al., How Powerful are Graph Neural Networks?, ICLR 2019

---

> ### Comment · Reviewer_1KdM · 2025-08-04
>
> Thank you for addressing my questions in the rebuttal.
>
> **W1)** Do I understand correctly that you agree that many widely-used GNN architectures already have the injectivity property for random initialization? In this case, the proposed architecture can not be considered a new contribution (as claimed), but only its analysis.
>
> **W2)** For a fair comparison of the dimensions, it would be necessary to distinguish the case where the $k$ is bounded (e.g., unlabeled graphs of bounded size) from the case where graphs can have real-valued node features. I do not agree with the advantage of your approach in terms of the required dimension suggested in the rebuttal.
>
> **W3)** I was wondering why you chose not to freeze the random parameters used in the aggregation step and claim that this is not problematic because of your novel architecture. As it seems, the architecture is not novel, and others (e.g., Morris et al., GIN) should have the same property. For me, it is unclear how to interpret the experimental results in light of this fact.
>
> **Minor comments**
>
> Please note that Corollary 8 in [1] applies to uncolored graphs only (or those with uniform node features). Using self-loops is generally not sufficient to reach WL expressivity, as shown, for example, in Foris Geerts et al. Let's Agree to Degree: Comparing Graph Convolutional Networks in the Message-Passing Framework. ICML 2021
> >It also helps explain why many expressivity-focused GNNs (e.g., higher-order variants) often show limited empirical gains: in practice, standard GNNs with sufficient width may already match 1-WL expressivity on real-world graphs. Our result thus highlights both the theoretical and practical boundaries of improving expressivity.
>
> I would disagree with this interpretation. Note that higher-order GNN are more expressive than 1-WL.

---

> > ### Author Response · Authors · 2025-08-04
> >
> > Thank you again for the follow-up comment. We would like to respectfully clarify a few points, as some of our earlier responses may have been misunderstood or overlooked:
> >
> > **W1) On Novelty of Analysis:** While we agree that many GNNs (e.g., GCN, GAT) are likely injective at random initialization, to our knowledge, this has **not previously been formally shown** before. With a small modification to the assumption (i.e., normalization preserving pairwise distinctness), our analysis can provably establish this result, which is a new and, we believe, important insight.
> >
> > Though structurally similar to prior GNNs, our 100-layer initialization experiments show that models like GIN and Morris et al. suffer from instability (NaNs, exploding norms) around 30 layers, while GCN, GAT, and SAGE oversmooth around 40. In contrast, our model maintains stable energy and full rank up to 100 layers. This distinction was not acknowledged in the follow-up.
> >
> > **W2): On Embedding Dimension ($k$ vs. $nd$):** Our analysis assumes uniform initial coloring, the worst case for structural separation, typically requiring more iterations to generate diverse embeddings. In practice, **real-valued node features** often stabilize **faster**, and do not worsen the bound. Our required width of $\tilde{\Omega}(k)$ already captures the worst case. Thus, our bound is not a best-case estimate, as the reviewer suggested, and it remains more efficient than the $\Omega(nd)$ bound in T. Amir et al., both in theory and in practice.
> >
> > **W3): On Freezing Aggregation Weights:** We explained that freezing aggregation layers can amplify random noise, harming downstream performance, which is why we train them. We also showed how our injectivity result could extend to the training regime using NTK and feature learning theory, and cited relevant work. This was restated as a question in the follow-up without engagement. As noted in W1, our 100-layer results also highlight a structural distinction that was not acknowledged.
> >
> > **Clarification on Uncolored Graphs and Self-Loops:**
> > As we noted earlier, uncolored graphs (or those with uniform initial features) require **more** iterations to stabilize WL colors, which is why our analysis focuses on this **harder** case. Our injectivity result depends on **summation-based aggregation** and **random nonlinear lifting**, not the self-loops. The citation to Geerts et al. (ICML 2021) focuses on **degree-oblivious GCNs**, and does not contradict our claim.
> >
> > **On the Role of Expressivity in Practice:**
> > We also want to emphasize that **injectivity is a sufficient but not necessary** condition for expressivity, and that many real-world benchmarks do not strongly challenge GNN expressivity. The table below compares the number of layers required to reach full WL coloring across models and datasets, as well as the number of unique node embeddings achieved:
> >
> > | Dataset   | #Nodes | Metric | WL     | GCN    | GAT    | GIN    | SAGE   | GraphConv | Ours   |
> > | --------- | ------ | ------ | ------ | ------ | ------ | ------ | ------ | --------- | ------ |
> > | Cora      | 2,708  | Layers | 6      | 4      | 5      | 5      | 8      | 5         | 6      |
> > |           |        | Embds  | 2,365  | 2,387  | 2,386  | 2,387  | 2,379  | 2,387     | 2,387  |
> > | Citeseer  | 3,327  | Layers | 7      | 5      | 6      | 6      | 9      | 6         | 7      |
> > |           |        | Embds  | 2,090  | 2,118  | 2,115  | 2,120  | 2,090  | 2,120     | 2,120  |
> > | Pubmed    | 19,717 | Layers | 6      | 4      | 5      | 5      | 7      | 5         | 6      |
> > |           |        | Embds  | 12,998 | 13,003 | 13,003 | 13,003 | 13,003 | 13,003    | 13,003 |
> > | Computers | 13,752 | Layers | 5      | 3      | 5      | 4      | 5      | 4         | 5      |
> > |           |        | Embds  | 13,349 | 13,357 | 13,356 | 13,358 | 13,355 | 13,358    | 13,358 |
> > | Photo     | 7,650  | Layers | 4      | 2      | 3      | 3      | 5      | 3         | 4      |
> > |           |        | Embds  | 7,460  | 7,465  | 7,464  | 7,466  | 7,465  | 7,466     | 7,466  |
> > | CS        | 18,333 | Layers | 6      | 4      | 5      | 5      | 6      | 5         | 6      |
> > |           |        | Embds  | 17,891 | 17,988 | 17,988 | 17,988 | 17,988 | 17,988    | 17,988 |
> > | Physics   | 34,493 | Layers | 6      | 4      | 5      | 5      | 6      | 8         | 6      |
> > |           |        | Embds  | 33,661 | 33,977 | 33,977 | 33,977 | 33,977 | 33,977    | 33,977 |
> > | WikiCS    | 11,701 | Layers | 5      | 2      | 3      | 4      | 4      | 3         | 3      |
> > |           |        | Embds  | 10,862 | 10,910 | 10,909 | 10,920 | 10,917 | 10,920    | 10,920 |
> >
> > Note: "Layers" $=$ WL iterations or GNN layers to reach the maximum unique colors or node embeddings. "Embds" $=$ distinct node representations after stabilization. Slight floating-point deviations can cause distinct node embeddings to exceed WL color count.

---

> > > ### Comment · Reviewer_1KdM · 2025-08-05
> > >
> > > Thanks for your reply. I respectfully disagree that my concerns are caused by overlooking arguments.
> > >
> > > **W1)**
> > > > While we agree that many GNNs (e.g., GCN, GAT) are likely injective at random initialization, to our knowledge, this has not previously been formally shown before.
> > >
> > > This is not correct. I have provided the corresponding literature in my review. The paper [2] states:
> > > > Using the fact that an embedding dimension of one is sufficient to achieve injectivity on $\mathcal{S}_{≤n}(\Omega)$ with countable $\Omega$, we show in Theorem 6.3 that standard MPNNs with analytic non-polynomial activations
> > > and random parameters have the separation power of WL, even when their architecture only uses a
> > > single feature per node.
> > >
> > > This result appears to be even stronger than yours (also see W2).
> > >
> > > Moreover, I have serious concerns regarding reinterpreting the contribution of your paper at this stage:
> > > Originally, you claimed that you proposed a new architecture having a unique injectivity property for random initialization. Your experiments show that it performs better than existing architectures for many layers. Now, it turns out that other architectures also have a similar injectivity property. So, a natural question is, why is the proposed architecture still better? Obviously, it is not the injectivity property as claimed. I appreciate the contribution of Proposition 2, but in my opinion, the motivation of the paper and the interpretation of the results need to be heavily adapted. Further investigations are necessary to explain the advantages of the proposed architecture. Moreover, its relation to existing architectures needs to be worked out.
> > >
> > > **W2)** For the injectivity property of a function $f\colon X\to Y$ realized by a neural network, the size (bounded, countable, uncountable) of the domain $X$ is crucial. This distinction has already been made in the seminal DeepSets paper. The quote from [2] presented above explicitly states that an embedding dimension of $1$ is sufficient in your setting with the existing approach (much better than near linear in $k$ you claim for your approach). The argument that graphs with uniform labels or node features are most difficult to distinguish is not relevant for this.
> > >
> > > **W3)** This point is connected to W1, and I have elaborated my concerns regarding the interpretation of the experimental results above. I fully understand the relation between layers, WL iterations, and initial features/labels.
> > >
> > >
> > > [2] T. Amir et al., Neural Injective Functions for Multisets, Measures and Graphs via a Finite Witness Theorem, NeurIPS 2023.

---

> > > > ### Author Response · Authors · 2025-08-06
> > > >
> > > > Thank you for the reviewer’s continued engagement. We would like to reinforce some clarifications, as certain points in our rebuttal appear to have been overlooked or misinterpreted.
> > > >
> > > > **W1):** We respectfully believe that the claim that *“many widely-used GNNs already have the injectivity property at random initialization”* requires further qualification. The paper [2] explicitly states that its result (Theorem 6.3) applies to **GIN-like MPNNs with summation aggregation** under certain assumptions. It does not at least directly cover architectures such as GCN, GAT, or GraphSAGE, which rely on **normalized or mean-based aggregation**—mechanisms that are known to reduce expressivity [6]. To our knowledge, [2] does not provide a formal injectivity result for these models, nor does it claim that mean-based aggregation shares the same expressive power as summation-based aggregation. If such results exist elsewhere, we would appreciate the reviewer sharing the relevant references.
> > > >
> > > > **W2):** While we agree that domain size influences theoretical width bounds, in practice, uniform initial features represent a worst-case scenario. These require more WL iterations—and thus deeper GNN layers—to resolve structural differences. This motivates our **decoupled design**, as discussed in Section 5.1 and Proposition 4: we apply shallow injective message passing to encode structure, then stop propagation to avoid oversmoothing. This addresses the gap between theoretical assumptions and practical needs.
> > > >
> > > > **W3):** We respectfully note that the reviewer did not engage with a central contribution in Section 5.2 and Lemma 1, which supports our 100-layer initialization experiments. Our results show that GIN and Morris et al. suffer from numerical instability (e.g., NaNs) by ~30 layers, and GCN, GAT, and GraphSAGE oversmooth around 50 layers. In contrast, our model maintains stable norm, Dirichlet energy, and full embedding rank beyond 50 layers — without residuals or normalization. This combination of provable injectivity and empirical depth stability provides a new perspective on GNN design, especially for decoupled architectures that separate feature learning from message passing.
> > > >
> > > > [2] T. Amir et al., Neural Injective Functions for Multisets, Measures and Graphs via a Finite Witness Theorem, NeurIPS 2023.
> > > >
> > > > [6] Xu, Keyulu, et al., How Powerful are Graph Neural Networks?, ICLR 2019

---

> > > > > ### Comment · Reviewer_1KdM · 2025-08-07
> > > > >
> > > > > I have no further questions. Just to clarify:
> > > > >
> > > > > **W1)** The paper [2] applies its result to a GIN-like GNN. Note that the analyzed architecture actually simply uses a one-layer MLP. The authors of [2] also refer to this architecture elsewhere as "standard MPNN", which is perfectly justified, as such an architecture is widely used. Considering that you also use a one-layer MLP in your submitted MPNN architecture, the results are closely related.
> > > > >
> > > > > [2] T. Amir et al., Neural Injective Functions for Multisets, Measures and Graphs via a Finite Witness Theorem, NeurIPS 2023.

---

> > > > > > ### Author Response · Authors · 2025-08-07
> > > > > >
> > > > > > Thanks for the discussion and engagement. However, we still wish to further clarify that the injectivity result in [2] relies on summation-based aggregation, whereas [6] (Figure 3, Corollary 8) provides counterexamples showing that mean or max aggregation—used in standard MPNNs such as GraphSAGE—may not always be injective. GCN and GAT further employ weighted-sum schemes without a formal injectivity guarantee. Therefore, it is not trivial to extend [2]’s result from GIN-like MPNNs to all “standard MPNNs.”
> > > > > >
> > > > > > [2] T. Amir et al., Neural Injective Functions for Multisets, Measures and Graphs via a Finite Witness Theorem, NeurIPS 2023.
> > > > > >
> > > > > > [6] K. Xu et al., How Powerful are Graph Neural Networks?, ICLR 2019.

---

### Official Review · Reviewer_JrE8 · 2025-07-04

**Clarity:** 4
**Significance:** 3
**Originality:** 3
**Rating:** 4
**Confidence:** 4

**Summary:**

The paper challenges the common belief that that increasing the depth of GNNs inherently leads to performance degradation, a phenomenon known as oversmoothing (the node embeddings become indistinguishable).

The authors argue that the root cause is the lack of injectivity in standard message passing (MP) mechanisms, which fail to preserve structural information across layers. To address this, the paper introduces a new message passing layer that is provably injective without requiring any training, ensuring that GNNs match the expressive power of the Weisfeiler-Lehman (WL) test by design. The proposed architecture decouples a shallow stack of injective MP layers (for structural expressivity) from a deep stack of feature learning layers (for representation learning).

Theoretical analysis is provided on the required depth, width, and initialization of MP layers to ensure both expressivity and numerical stability.

Empirical results on standard node classification benchmarks demonstrate that the proposed approach enables deeper GNNs without suffering from oversmoothing, offering a new perspective on building scalable and expressive GNNs.

**Questions:**

I appreciate authors thoughts on the followings in addition to weaknesses mentioned before:

- How can injective message passing be integrated with node sampling strategies to address the problem of neighborhood explosion?

- Can the proposed injective MP and decoupled architecture be effectively applied to graph-level tasks (e.g., graph classification) or link prediction, and how would the design need to be adapted?

**Ethical Concerns:**

["NO or VERY MINOR ethics concerns only"]

**Final Justification:**

I am grateful for the authors’ thoughtful response. Taking authors' feedback and comments from other reviewers into account, I lean toward keeping my score unchanged

**Limitations:**

Please see the weaknesses above.

**Quality:**

3

**Strengths And Weaknesses:**

The authors provide a rigorous theoretical framework showing that injectivity, not depth, is the key to avoiding oversmoothing in GNNs, which is an interesting observation.  The introduction of injective message passing layer, with formal proofs and clear assumptions is also novel. The proofs  in appendix,  while building on known results, looks sound to me as far as I checked. On the empirical side, the comprehensive experiments were conducted on eight node classification datasets, with results showing competitive or superior performance to strong GNN baselines, where authors include ablation studies on the effects of MP width, depth, and training, supporting the theoretical claims. I also appreciate authors providing details on datasets, hyperparameters, training protocols, and compute resources are provided, supporting reproducibility.

However, I found the following issues that may need further consideration:
- The condition in Proposition 1 where all the node embeddings are linearly independent, needs further elaboration. While this makes sense as it is known that oversmoothing can be naturally explained by the the fact that the rank of embedding matrix becomes smaller, but in real application the embedding matrix might be low rank and not full rank as indicated by this condition.
- It would be nice if authors put their results in context of prior works that also challenge the oversommething, e.g., On provable benefits of depth in training graph convolutional networks, NeurIPS 2021
- The experiments are limited to node classification tasks on standard citation and co-authorship datasets; generalization to other graph tasks (e.g., graph classification, link prediction) or domains (e.g., molecular graphs, social networks) is not explored in depth. The impact of the proposed method on very large-scale or highly heterogeneous graphs is not extensively analyzed.

---

> ### Author Rebuttal · Authors · 2025-07-28
>
> **1. Clarifying Proposition 1 and the full-rank assumption**:
> Thank you for raising this point. We would like to clarify that Proposition 1 only requires the embeddings of **distinct** nodes to be linearly independent—this is weaker than requiring the full embedding matrix to be full-rank. Thus, certain low-rank cases are already covered.
>
> Moreover, our theoretical analysis focuses on the **worst-case scenario**, aiming to provide an upper bound on the number of layers needed to match WL expressivity. This is why Theorem 1 assumes identical input features (i.e., rank 1)—a minimal and challenging setting. In practice, higher-rank or more diverse features typically require **fewer** layers for expressivity. We will revise the manuscript to better contextualize this point.
>
> **2. Relation to Depth and Oversmoothing Literature**: We appreciate the reviewer for highlighting the NeurIPS 2021 paper “On the Provable Benefits of Depth in Training GCNs.” This work aligns with our perspective that depth, by itself, is not the root cause of performance degradation in GNNs.
>
> Their theoretical results show that deeper GCNs improve expressivity and maintain good training accuracy, and that the generalization gap—not oversmoothing—is often the limiting factor. Our work complements this view by identifying lack of injectivity as another bottleneck in deep GNNs. Specifically, we provide theoretical and empirical evidence that GNNs with injective message passing (even when frozen) preserve expressivity and avoid collapse.
>
> Both papers advocate for architectural decoupling as a means to harness depth more effectively. While the NeurIPS paper decouples weights from depth via residual reparameterization, we decouple message passing from feature learning—using shallow injective MP layers to capture structure and deep MLPs for representation learning. We believe this shared motivation reinforces the value of decoupling-based strategies for improving deep GNN performance. We will include this discussion and cite the paper in our revised manuscript.
>
> **3. Compatibility with Sampling-Based GNN Training:** Thank you for the thoughtful question. Our injective message passing mechanism naturally extends to mini-batch training with sampling strategies, including layer-wise neighborhood sampling (e.g., GraphSAGE) and subgraph sampling (e.g., GraphSAINT, Cluster-GCN). In all cases, message passing operates over a valid subgraph of the original graph.
>
> Our injectivity guarantee relies on the condition that the (distinct) node embeddings of the sampled neighbors are **linearly independent**, which is enforced via a **nonlinear lifting** step (e.g., a sufficiently wide MLP). This holds whether the neighbors are sampled from the full graph or from a smaller subset. In fact, sampling often results in fewer neighbors, making injectivity easier to satisfy. Thus, our method is both injective and scalable.
>
> **4. Limited task diversity and scale**: Thank you for raising the concern regarding the scope of our experiments. We have now added **graph-level classification** results on five standard datasets (e.g., MUTAG, NCI1, PROTEINS, REDDIT-B), showing that our method achieves competitive or superior performance across diverse domains as below. While we have not yet included link prediction or heterophilous graphs due to space and resource constraints in the rebuttal, we agree that these are important directions and plan to explore them and add the results in the revised paper.
>
> > We use the same tuning strategy as for node classification in the paper. Bolded values indicate the best performance per dataset.
>
> | Model        | MUTAG           | PTC             | NCI1            | PROTEINS         | REDDIT-B         |
> |--------------|------------------|------------------|------------------|------------------|------------------|
> | # Graphs     | 188              | 344              | 4110             | 1113             | 1000             |
> | # Classes    | 2                | 2                | 2                | 2                | 2                |
> | Avg. # Nodes | 17.9             | 25.5             | 29.8             | 39.1             | 19.8             |
> | **GCN**      | 0.9257 ± 0.06    | 0.7850 ± 0.04    | **0.8474 ± 0.01**| 0.7943 ± 0.02    | 0.9130 ± 0.02    |
> | **SAGE**     | 0.9099 ± 0.07    | 0.7707 ± 0.06    | 0.8409 ± 0.01    | 0.7898 ± 0.03    | 0.7625 ± 0.02    |
> | **GAT**      | 0.9260 ± 0.07    | 0.7939 ± 0.04    | 0.8345 ± 0.01    | 0.7853 ± 0.02    | 0.7896 ± 0.03    |
> | **GIN**      | 0.9523 ± 0.06    | 0.7641 ± 0.05    | 0.8431 ± 0.01    | 0.7763 ± 0.02    | 0.9190 ± 0.01    |
> | **GraphConv**| 0.9471 ± 0.06    | 0.7618 ± 0.06    | 0.8494 ± 0.02    | 0.7826 ± 0.03    | 0.9015 ± 0.01    |
> | **Ours**     | **0.9576 ± 0.06**| **0.8082 ± 0.02**| 0.8380 ± 0.01    | **0.7997 ± 0.02**| **0.9265 ± 0.02**|
>
>
> To further broaden our empirical evaluation, we also include new oversmoothing mitigation experiments across three node classification benchmarks—small (Cora), medium (Pubmed), and **large** (OGBN-Arxiv)—to evaluate the effect of widely used techniques such as residual connections and normalization to our method. Interestingly, our results show that while these techniques offer marginal improvements on standard GNNs, our method already achieves strong performance without requiring additional architectural tweaks. These findings provide a more comprehensive view of how our approach scales and generalizes across different tasks and graph regimes.
>
> | Model         | Cora (2.7K)      | Pubmed (19.7K)   | OGBN-Arxiv (169K) |
> | ------------- | ---------------- | ---------------- | ----------------- |
> | **GCN**       | 83.16 ± 1.14     | 78.90 ± 0.93     | 69.92 ± 0.01      |
> | + Norm        | 80.64 ± 0.26     | 77.32 ± 0.66     | **72.74 ± 0.05**  |
> | + Res         | 82.48 ± 1.64     | 77.50 ± 0.83     | 71.80 ± 0.06      |
> | **SAGE**      | 80.56 ± 1.34     | 77.10 ± 0.60     | 70.78 ± 0.02      |
> | + Norm        | 76.12 ± 2.61     | 74.26 ± 1.12     | 72.50 ± 0.07      |
> | + Res         | 80.68 ± 0.73     | 76.84 ± 1.69     | 71.04 ± 0.06      |
> | **GAT**       | 81.92 ± 1.33     | 77.02 ± 1.13     | 69.93 ± 0.06      |
> | + Norm        | 78.70 ± 0.23     | 76.30 ± 1.30     | 72.15 ± 0.02      |
> | + Res         | 83.06 ± 0.77     | 76.38 ± 0.83     | 71.10 ± 0.13      |
> | **GIN**       | 79.96 ± 1.42     | 75.14 ± 0.90     | 36.32 ± 0.38      |
> | + Norm        | 69.82 ± 3.62     | 74.82 ± 1.26     | 70.86 ± 0.26      |
> | + Res         | 80.30 ± 0.97     | 72.88 ± 3.12     | 47.13 ± 2.55      |
> | **GraphConv** | 82.08 ± 1.09     | 76.40 ± 0.76     | 69.52 ± 0.15      |
> | + Norm        | 76.40 ± 1.45     | 74.06 ± 2.02     | 72.60 ± 0.25      |
> | + Res         | 81.58 ± 0.31     | 75.96 ± 1.20     | 70.96 ± 0.13      |
> | **Ours**      | **83.74 ± 0.55** | **79.16 ± 1.23** | 72.71 ± 0.02      |
> | + Norm        | 75.28 ± 2.24     | 76.81 ± 0.83     | **72.73 ± 0.06**  |
> | + Res         | 83.38 ± 1.53     | **79.56 ± 1.28** | 72.72 ± 0.38      |
> > To prevent overfitting, all GNNs are tuned with dropout. Bolded values indicate the best performance per dataset.

---

### Official Review · Reviewer_Uczg · 2025-07-18

**Clarity:** 2
**Significance:** 3
**Originality:** 2
**Rating:** 4
**Confidence:** 3

**Summary:**

The authors propose a variant of the standard sum‑aggregation message‑passing GNN in which the initial layers use a fixed random neural network for message passing. This architectural choice ensures (almost sure) injectivity of the initial layers, preserving the model’s expressiveness to match exactly that of the 1‑WL test. The paper suggests appropriate network depths by extrapolating from the depths required to distinguish binary trees and 2D grids. Experiments compare the new model with state‑of‑the‑art GNNs, showing competitive performance, and additionally examine empirically how performance depends on GNN depth and width.

**Questions:**

1. The authors note that standard sum‑aggregation GNNs can lose expressivity during training by breaking injectivity. But their own model also suffers this: although the frozen layers remain injective, the subsequent (trainable) layers may not. What, then, is the practical benefit of the proposed architecture over standard sum‑aggregation GNNs?

2. The central claim appears to be that depth doesn’t harm GNNs—loss of injectivity does. Yet there is little empirical evidence that the proposed model outperforms standard models at large depths. How do the experiments substantiate this claim? The abstract’s statement—“Empirically, we demonstrate that our architecture enables deeper GNNs without suffering from oversmoothing”—is not clearly supported by the presented results.

3. In the depth experiments, the initially frozen MLPs are retrained. In that setting, how does the proposed architecture differ from a standard GNN? It seems the differences become negligible once those layers are trainable.

**Ethical Concerns:**

["NO or VERY MINOR ethics concerns only"]

**Final Justification:**

I have increased my rating for two reasons. First, the additional experiments support the papers main claim. Second, the connection with [1] suggested by the AC has lead to discussion which greatly clears up the significance and motivation of the paper. The authors have agreed to include some of this discussion.

**Limitations:**

Yes

**Paper Formatting Concerns:**

Figures 1 and 2 use fonts that are extremely small.

**Quality:**

2

**Strengths And Weaknesses:**

**Strengths**

* Addresses an important challenge in GNN design with both empirical and theoretical motivation.
* Experiments demonstrate modest improvements over state‑of‑the‑art models.

**Weaknesses**

* Clarity is a significant issue, making some results hard to follow.
* The theoretical advantages appear minimal: although the frozen random layers are injective, the overall GNN need not be (although it’s not obvious that full injectivity is even desirable).
* Expressivity is characterized only in terms of graph‑distinguishing power; it remains unclear which classes of graph functions the model can implement. Freezing parameters may in fact restrict expressivity.

---

> ### Author Rebuttal · Authors · 2025-07-27
>
> **1. Injectivity is broken after frozen layers, then what is the benefit?** We thank the reviewer for raising this concern. While it's true that injectivity could, in principle, be lost after the frozen message passing layers, prior work on NTK theory [1,2] and feature learning theory [3] suggests otherwise: embeddings in pure MLPs tend to remain linearly independent throughout training. This implies that injectivity likely holds for the subsequent (trainable) layers, even during training. We also plan to build on these insights to extend our current initialization-time results to the full training dynamics.
>
> **2. Empirical evidence for depth claim is limited**: We thank the reviewer’s concern and agree our original results did not fully substantiate the claim. During the rebuttal, we conducted **100-layer initialization experiments** comparing GIN, GraphConv, GCN, GAT, SAGE, and our method, all **without skip connections, normalization, or dropout**, to isolate the effects of pure message passing. Our findings show:
>
> - **GCN, GAT, and SAGE** suffer from **oversmoothing**, with Dirichlet energy and embedding norm decay, and embedding rank collapses to zero by ~50 layers.
> - **GIN and GraphConv** exhibit severe **numerical instability**, with exploding energy and norms, yielding `inf/NaN` values as early as 30 layers.
> - **Our method** maintains stable energy, norm, and full rank beyond 50 layers, degrading gracefully even at depth 100.
>
> Due to rebuttal response constraints, we report  Dirichlet energy and embedding rank on the Cora dataset as robust metrics here.  Full results on other datasets will be included in the revised paper. These results clarify that oversmoothing and instability are common failure modes in deep GNNs, and our method is significantly more stable at large depths.
>
> | Layer | Metric  | GCN    | GAT    | GIN   | SAGE   | GraphConv | Ours   |
> | ----- | ------ | ------ | ------ | ----- | ------ | --------- | ------ |
> | 001   | log(E) | 3.48   | 5.15   | 7.21  | 4.67   | 9.61      | 5.96   |
> |       | Rank   | 100    | 36     | 37    | 33     | 37        | 37     |
> | 020   | log(E) | -28.93 | -20.81 | 38.72 | -29.33 | 73.84     | -7.17  |
> |       | Rank   | 100    | 100    | 100   | 100    | 100       | 100    |
> | 050   | log(E) | -72.60 | -62.29 | inf   | -83.61 | inf       | -29.31 |
> |       | Rank   | 1      | 1      | NaN   | NaN      | NaN       | 100    |
> | 100   | log(E) | -inf   | -inf   | inf   | -inf   | inf       | -63.46 |
> |       | Rank   | NaN      | NaN      | NaN   | NaN      | NaN       | 1      |
> *(log(E) = log Dirichlet energy; Rank = embedding rank)*
>
> **3. No clear empirical advantage over other GNNs:**: We agree that our method does not always outperform other GNNs. In fact, on common benchmarks, many GNNs perform similarly when well-tuned. To better understand this, we conducted additional experiments measuring the **expressivity** of GNNs using the number of distinct node embeddings as a proxy—aligned with the WL test's distinct colorings, following [4–5]. Our key findings are:
>
> - **Most GNNs match WL-level expressivity within 2–6 layers**, even with modest width. This supports the idea that *injectivity is sufficient but not always necessary*, helping explain why improving expressivity does not always yield performance gains.
> - **~90% of nodes have distinct WL colors after just a few iterations**, indicating high structural uniqueness. In such settings, deeper message passing may amplify noise rather than help, potentially harming generalization.
>
> These observations explain why accuracy remains flat across many architectures and support our **decoupled** design. Since expressivity saturates very early, only shallow message passing is needed to capture structural information. Meanwhile, oversmoothing tends to occur much later (beyond ~20 layers), so it can be avoided by switching early to deeper MLPs for feature learning—preserving expressivity while enabling deeper networks without degradation.
>
> | Dataset           | #Nodes   | Metric   | WL     | GCN    | GAT    | GIN    | SAGE   | GraphConv | Ours   |
> |------------------|----------|----------|--------|--------|--------|--------|--------|------------|--------|
> | **Cora**         | 2,708    | Layers   | 6      | 4      | 5      | 5      | 8      | 5          | 6      |
> |                  |          | Embds    | 2,365  | 2,387  | 2,386  | 2,387  | 2,379  | 2,387      | 2,387  |
> | **Citeseer**     | 3,327    | Layers   | 7      | 5      | 6      | 6      | 9      | 6          | 7      |
> |                  |          | Embds    | 2,090  | 2,118  | 2,115  | 2,120  | 2,090  | 2,120      | 2,120  |
> | **Pubmed**       | 19,717   | Layers   | 6      | 4      | 5      | 5      | 7      | 5          | 6      |
> |                  |          | Embds    | 12,998 | 13,003 | 13,003 | 13,003 | 13,003 | 13,003     | 13,003 |
> | **Computers**    | 13,752   | Layers   | 5      | 3      | 5      | 4      | 5      | 4          | 5      |
> |                  |          | Embds    | 13,349 | 13,357 | 13,356 | 13,358 | 13,355 | 13,358     | 13,358 |
> | **Photo**        | 7,650    | Layers   | 4      | 2      | 3      | 3      | 5      | 3          | 4      |
> |                  |          | Embds    | 7,460  | 7,465  | 7,464  | 7,466  | 7,465  | 7,466      | 7,466  |
> | **CS**           | 18,333   | Layers   | 6      | 4      | 5      | 5      | 6      | 5          | 6      |
> |                  |          | Embds    | 17,891 | 17,988 | 17,988 | 17,988 | 17,988 | 17,988     | 17,988 |
> | **Physics**      | 34,493   | Layers   | 6      | 4      | 5      | 5      | 6      | 8          | 6      |
> |                  |          | Embds    | 33,661 | 33,977 | 33,977 | 33,977 | 33,977 | 33,977     | 33,977 |
> | **WikiCS**       | 11,701   | Layers   | 5      | 2      | 3      | 4      | 4      | 3          | 3      |
> |                  |          | Embds    | 10,862 | 10,910 | 10,909 | 10,920 | 10,917 | 10,920     | 10,920 |
>
> > Note: "Layers" refers to the number of WL iterations or GNN layers required to reach the maximum number of distinct node embeddings. "Embds" refers to the number of unique node representations after stabilization. Slight floating-point deviations can cause distinct node embeddings to exceed WL color count.
>
> These findings suggest that existing benchmarks do not always challenge GNN expressivity significantly—most architectures already match the WL test early on. This raises new questions for GNN design, especially in *over-unique regimes*, and we see this as a valuable direction for future work.
>
> These results also clarify why improving injectivity does not always lead to better accuracy on current benchmarks: most nodes already have unique structures, so expressivity saturates quickly. This over-uniqueness makes deeper message passing unnecessary—and even harmful—by amplifying noise. It reinforces our decoupled design and points to future work on characterizing **over-uniqueness** and designing benchmarks where deeper structural reasoning is essential.
>
> **4. On frozen vs. trainable layers**: We agree that training the initially frozen message passing layers improves performance, and on standard benchmarks, our method does not always outperform other GNNs. As discussed, *injectivity is sufficient but not always necessary* and most datasets does not challenge GNN expressivity. Moreover, our observation of over-uniqueness—where most nodes have distinct structures early—suggests that deeper message passing may be unnecessary or even harmful. This reinforces the value of our decoupled design: using shallow injective message passing to capture structure, followed by deeper MLPs for feature learning, helps avoid both over-uniqueness and oversmoothing.
>
> **5. Clarity Issue:** We appreciate the reviewer’s time and feedback. Regarding the clarity concern, we respectfully note that all other reviewers rated the paper’s clarity as good or excellent. It is unclear to us which parts were confusing, as the paper includes both theoretical analysis and empirical results, following standard presentation practices in the GNN literature. In this rebuttal, we have provided additional empirical evidence and detailed explanations, which we hope have clarified our contributions. We would be happy to further revise and improve the paper if the reviewer could indicate which parts were difficult to follow.
>
> **6. Expressivity Framework:** On expressivity, we agree that function approximation is a rich and important direction, especially in broader deep learning. However, in the GNN community, it is common and well-established to evaluate expressivity through graph distinguishing power, particularly via the WL test. This framing has been widely adopted since the foundational works by [4-5], and our analysis is aligned with this established theoretical framework. On the other hand, the trainable MLP layers after message passing provide expressive power from a function approximation standpoint—even when the message passing layers are frozen. We welcome broader discussions on alternative expressivity metrics, but we believe our approach is consistent with current standards and offers novel insight into both the role of injectivity and the limitations of depth in real-world GNN benchmarks.
>
> [1] Jacot, Arthur, et al., Neural tangent kernel: Convergence and generalization in neural networks, NeurIPS 2018
>
> [2] Du, Simon, et al., Gradient descent finds global minima of deep neural networks, ICML 2019.
>
> [3] Chen, Zixiang, et al., Global Convergence and Rich Feature Learning in $L$-Layer Infinite-Width Neural Networks under $\mu P$ Parametrization, ICML 2025
>
> [4] C. Morris et al., Weisfeiler and Leman Go Neural: Higher-order Graph Neural Networks, AAAI 2019.
>
> [5] Xu, Keyulu, et al., How Powerful are Graph Neural Networks?, ICLR 2019

---

> > ### Comment · Reviewer_Uczg · 2025-08-02
> >
> > Thank you for the additional empirical results which do seem to provide evidence supporting the central claim on depth.
> >
> > It appears that other reviewers found the paper much easier to understand, so my comment on clarity should be taken lightly. To be more specific, I find all the technical content suitably precise. What is hard for me to understand is the motivation for the proposed approach. In particular, it is very non-intuitive for these injective layers to be useful if they are not trainable. This is exactly my concern with expressivity. The frozen layers are expressive in terms of isomorphism testing, but they cannot be expressive in terms of function approximation by nature of being frozen.
> >
> > A relevant paper [1] has been brought to my attention, and the results here are very useful in helping me understand the usefulness of your approach. In particular, I believe Theorem 2 of [1] shows that appending a trainable NN layer to your frozen is sufficient to universal function approximation. I think the clarity would greatly benefit from making this connection explicit.
> >
> > [1] Chen, Z., Villar, S., Chen, L., & Bruna, J. (2019). On the equivalence between graph isomorphism testing and function approximation with gnns. Advances in neural information processing systems.

---

> > > ### Author Response · Authors · 2025-08-03
> > >
> > > We sincerely thank the reviewer for their thoughtful post-rebuttal response and for acknowledging that our additional experiments support the central claim regarding depth. We are also grateful for the reference to [1], which we found highly relevant and illuminating.
> > >
> > > Upon studying [1], we found that it helps bridge the gap between graph isomorphism testing and function approximation in GNNs. Specifically, Theorem 2 (and its extension to compact domains in Theorem 4) shows that a collection of expressive, permutation-invariant functions (e.g., WL-expressive GNNs) followed by a trainable neural network (e.g., MLP) is sufficient for universal function approximation.
> > >
> > > This insight directly supports our decoupled architecture: the frozen injective message passing layers extract structural information aligned with WL-test expressivity, while subsequent trainable MLPs perform flexible representation learning. It also addresses the earlier concern: although the frozen layers themselves are not functionally expressive in isolation, the **overall architecture becomes universal** when paired with downstream MLPs.
> > >
> > > Moreover, this connection suggests a theoretical advantage of our design: performing structural encoding **before** feature learning is a principled and theoretically justified strategy—complementary to alternative designs like APPNP [2], which reverse this order. We plan to include this discussion in the revised paper, adding a paragraph along the following lines:
> > >
> > > > While our frozen injective message passing layers are designed to encode structural information (i.e., capture graph isomorphism classes), their lack of trainability raises natural concerns about functional expressivity. To address this, we draw on insights from [1], which show that appending a trainable MLP to an expressive GNN is sufficient for universal function approximation. Our decoupled design follows this principle: we first encode graph structure through injective (frozen) MP layers, then apply trainable MLPs for task-specific learning—achieving both structural expressivity and functional flexibility. Notably, this also supports the idea that encoding structure prior to feature learning is a theoretically grounded design choice, in contrast to approaches like APPNP [2], which apply message passing after MLP-based transformations.
> > >
> > > [1] Chen, Z., Villar, S., Chen, L., & Bruna, J. (2019). On the equivalence between graph isomorphism testing and function approximation with GNNs. Advances in Neural Information Processing Systems.
> > >
> > > [2] Gasteiger, Johannes, et al. (2019). Predict then Propagate: Graph Neural Networks meet Personalized PageRank. ICLR.

---

### Official Review · Reviewer_unzM · 2025-07-22

**Clarity:** 3
**Significance:** 3
**Originality:** 3
**Rating:** 4
**Confidence:** 2

**Summary:**

The paper is interested in studying the effect of depth of the message-passing framework on the performance of the model. The authors shows that the effect of depth, which is commonly connected to oversmoothing, is actually more connected to the injectivity of the message-passing operation. In this perspective, they propose a new message-passing scheme that is injective and match the expressive power of WL test.

**Questions:**

The main questions are related to the weakness section (please refer to that part for more clarity and context):
- Can you provided additional experimental baselines (specifically methods that were meant to address the over-smoothing effect).
- Can you additionally provide results on larger datasets ?

**Ethical Concerns:**

["NO or VERY MINOR ethics concerns only"]

**Final Justification:**

While the authors have provided some elements in their rebuttal around some of the questions I had, I still think that a set of additional experiments comparing to advanced methods that tackle over-smoothing are missing.

In addition, after carefully reading the other reviewer's discussion and point of view, I think some points raised by reviewer 1KdM are valid and the novelty of the work is questionable. Nonetheless, I would like to point here that I am not very knowledgeable in this specific area of GNNs (reflected by my confidence of 2) and therefore while I lean towards a "borderline accept", I believe some of the point raised by Reviewer 1KdM, and therefore I don't have a *strong* objection on the rejection of the paper and would rather rely on the other reviewers and the AC in this aspect.

**Limitations:**

Please refer to the Weaknesses section. I believe the main limitation of the paper is the experimental setup and the limited considered baselines and datasets.

**Paper Formatting Concerns:**

Only small elements to report here, mainly:
- The figures are not very clear and therefore putting rather larger captions and legend would make reading the paper easier.
- In Section E in the Appendix, you state that additional datasets (such as CS and Photo) will be added in the supplementary materials which are actually added to the main paper.

**Quality:**

3

**Strengths And Weaknesses:**

**Strength:**
- The authors approached the problem from a new perspective, giving therefore new insights different from those previously available which are mainly connected to the oversmoothing.
- The paper is very well written and clear. Specifically, the introduction of the independence (through Proposition 2) is very useful to understand the proposed adaptation of the message-passing proposed by the authors.
- The authors provided theoretical elements studying their proposed scheme from an expressivity point of view (Theorem 1).
- I think the proposed decoupled architecture is very interesting and provides a new direction.

**Weaknesses:**

I think the main weakness of this paper is within its experimental setup. Specifically I can summarise my comments as follows:
- The proposed baselines (GCN, SAGE and GAT) are very limited, and the authors have chosen to omit a number of sophisticated GNN architecture that specifically address the problem of oversmoothing (which were actually discussed in the related work section). This lack of comparison makes it therefore difficult to assess the performance of the proposed message-passing.
- The considered datasets are relatively small and therefore I am wondering about the performance of the method on larger datasets (such as the OGB graphs). Specifically, such experiment shall give the light on the method’s ability to distinguish two different nodes (based on the 1-WL) as discussed  Proposition 4.
- In my opinion, the method’s performance compared to the other baselines is very marginal specifically when comparing the author’s proposed method and the GCN (specifically when taking into account the standard deviation).
- While from my understanding, the proposed paper theoretically reduces the complexity of the message-passing scheme compared to the other variants, no empirical analysis of such gain is provided in the paper.

---

> ### Author Rebuttal · Authors · 2025-07-29
>
> **1. Oversmoothing experiments**: We appreciate the reviewer raising this important concern. To address it, we conducted additional experiments during the rebuttal phase:
>
> **(a) 100-layer initialization analysis:** We conducted controlled experiments on GCN, GAT, GIN, SAGE, GraphConv, and our method—all **without skip connections, normalization, or dropout**—to isolate oversmoothing effects. We observed that GCN, GAT, and SAGE suffer from oversmoothing: their Dirichlet energy decay and rank collapse by ~50 layers. GIN and GraphConv experience severe instability, producing `inf` or `NaN` values as early as 30–40 layers. Our method, by contrast, maintains stable energy, bounded norms, and full embedding rank up to depth 50, and degrades gracefully even at depth 100. We report **Dirichlet energy** and **embedding rank** here as robust metrics:
>
> | Layer | Metric  | GCN    | GAT    | GIN   | SAGE   | GraphConv | Ours   |
> | ----- | ------ | ------ | ------ | ----- | ------ | --------- | ------ |
> | 001   | log(E) | 3.48   | 5.15   | 7.21  | 4.67   | 9.61      | 5.96   |
> |       | Rank   | 100    | 36     | 37    | 33     | 37        | 37     |
> | 020   | log(E) | -28.93 | -20.81 | 38.72 | -29.33 | 73.84     | -7.17  |
> |       | Rank   | 100    | 100    | 100   | 100    | 100       | 100    |
> | 050   | log(E) | -72.60 | -62.29 | inf   | -83.61 | inf       | -29.31 |
> |       | Rank   | 1      | 1      | NaN   | NaN      | NaN       | 100    |
> | 100   | log(E) | -inf   | -inf   | inf   | -inf   | inf       | -63.46 |
> |       | Rank   | NaN      | NaN      | NaN   | NaN      | NaN       | 1      |
> *(log(E) = log Dirichlet energy; Rank = embedding rank)*
>
> These results show that oversmoothing and instability are prevalent failure modes in deep GNNs, and our method exhibits substantially greater stability.
>
> **(b) Impact of Oversmoothing Mitigation Techniques:** Prior theoretical work [1] and recent benchmarking studies [2] have shown that residual connections and normalization layers can mitigate oversmoothing and significantly improve the performance of classical GNNs. To evaluate this effect, we tested these techniques on GCN, SAGE, GAT, GIN, GraphConv, and our method across three node classification benchmarks: Cora (small), Pubmed (medium), and OGBN-Arxiv (large).
>
> While these techniques do help reduce oversmoothing in baseline models, our method already achieves strong performance without them and is only marginally improved when they are added. This suggests that our injective message passing inherently addresses depth-related degradation, without the need for architectural add-ons.
>
> > To prevent overfitting, all GNNs are tuned with dropout. Bolded values indicate the best performance per dataset.
>
> | Model         | Cora (2.7K)      | Pubmed (19.7K)   | OGBN-Arxiv (169K) |
> | ------------- | ---------------- | ---------------- | ----------------- |
> | **GCN**       | 83.16 ± 1.14     | 78.90 ± 0.93     | 69.92 ± 0.01      |
> | + Norm        | 80.64 ± 0.26     | 77.32 ± 0.66     | **72.74 ± 0.05**  |
> | + Res         | 82.48 ± 1.64     | 77.50 ± 0.83     | 71.80 ± 0.06      |
> | **SAGE**      | 80.56 ± 1.34     | 77.10 ± 0.60     | 70.78 ± 0.02      |
> | + Norm        | 76.12 ± 2.61     | 74.26 ± 1.12     | 72.50 ± 0.07      |
> | + Res         | 80.68 ± 0.73     | 76.84 ± 1.69     | 71.04 ± 0.06      |
> | **GAT**       | 81.92 ± 1.33     | 77.02 ± 1.13     | 69.93 ± 0.06      |
> | + Norm        | 78.70 ± 0.23     | 76.30 ± 1.30     | 72.15 ± 0.02      |
> | + Res         | 83.06 ± 0.77     | 76.38 ± 0.83     | 71.10 ± 0.13      |
> | **GIN**       | 79.96 ± 1.42     | 75.14 ± 0.90     | 36.32 ± 0.38      |
> | + Norm        | 69.82 ± 3.62     | 74.82 ± 1.26     | 70.86 ± 0.26      |
> | + Res         | 80.30 ± 0.97     | 72.88 ± 3.12     | 47.13 ± 2.55      |
> | **GraphConv** | 82.08 ± 1.09     | 76.40 ± 0.76     | 69.52 ± 0.15      |
> | + Norm        | 76.40 ± 1.45     | 74.06 ± 2.02     | 72.60 ± 0.25      |
> | + Res         | 81.58 ± 0.31     | 75.96 ± 1.20     | 70.96 ± 0.13      |
> | **Ours**      | **83.74 ± 0.55** | **79.16 ± 1.23** | 72.71 ± 0.02      |
> | + Norm        | 75.28 ± 2.24     | 76.81 ± 0.83     | **72.73 ± 0.06**  |
> | + Res         | 83.38 ± 1.53     | **79.56 ± 1.28** | 72.72 ± 0.38      |
>
> **(c) Comparisons to advanced oversmoothing-focused architectures:** Due to time and space constraints during the rebuttal phase, we were unable to include experiments with more complex models (e.g., Graph-Coupled Oscillator Network and Gradient Gating). We agree these are valuable baselines, and we will add comparative experiments in the revised paper.
>
> **2. Larger-scale datasets (OGBN-Arxiv):** We added results on **OGBN-Arxiv (169K nodes, 1.1M edges)** to evaluate scalability. Our method maintains strong performance even without using oversmoothing techniques. To further study depth and expressivity, we analyzed the number of layers needed by each model to match the number of distinct colors produced by the 1-WL test across eight datasets. Surprisingly, we found that most benchmarks require only ~5 WL iterations for full neighborhood separation, meaning even shallow GNNs can match 1-WL expressivity. This helps explain the limited performance differences and suggests that deeper propagation is often unnecessary on standard benchmarks. We are exploring more challenging datasets where depth plays a greater role and will include those results in the final version.
>
> > Note: "Layers" refers to the number of WL iterations or GNN layers required to reach the maximum number of distinct node embeddings. "Embds" refers to the number of unique node representations after stabilization. Slight floating-point deviations can cause distinct node embeddings to exceed WL color count.
> | Dataset           | #Nodes   | Metric   | WL     | GCN    | GAT    | GIN    | SAGE   | GraphConv | Ours   |
> |------------------|----------|----------|--------|--------|--------|--------|--------|------------|--------|
> | **Cora**         | 2,708    | Layers   | 6      | 4      | 5      | 5      | 8      | 5          | 6      |
> |                  |          | Embds    | 2,365  | 2,387  | 2,386  | 2,387  | 2,379  | 2,387      | 2,387  |
> | **Citeseer**     | 3,327    | Layers   | 7      | 5      | 6      | 6      | 9      | 6          | 7      |
> |                  |          | Embds    | 2,090  | 2,118  | 2,115  | 2,120  | 2,090  | 2,120      | 2,120  |
> | **Pubmed**       | 19,717   | Layers   | 6      | 4      | 5      | 5      | 7      | 5          | 6      |
> |                  |          | Embds    | 12,998 | 13,003 | 13,003 | 13,003 | 13,003 | 13,003     | 13,003 |
> | **Computers**    | 13,752   | Layers   | 5      | 3      | 5      | 4      | 5      | 4          | 5      |
> |                  |          | Embds    | 13,349 | 13,357 | 13,356 | 13,358 | 13,355 | 13,358     | 13,358 |
> | **Photo**        | 7,650    | Layers   | 4      | 2      | 3      | 3      | 5      | 3          | 4      |
> |                  |          | Embds    | 7,460  | 7,465  | 7,464  | 7,466  | 7,465  | 7,466      | 7,466  |
> | **CS**           | 18,333   | Layers   | 6      | 4      | 5      | 5      | 6      | 5          | 6      |
> |                  |          | Embds    | 17,891 | 17,988 | 17,988 | 17,988 | 17,988 | 17,988     | 17,988 |
> | **Physics**      | 34,493   | Layers   | 6      | 4      | 5      | 5      | 6      | 8          | 6      |
> |                  |          | Embds    | 33,661 | 33,977 | 33,977 | 33,977 | 33,977 | 33,977     | 33,977 |
> | **WikiCS**       | 11,701   | Layers   | 5      | 2      | 3      | 4      | 4      | 3          | 3      |
> |                  |          | Embds    | 10,862 | 10,910 | 10,909 | 10,920 | 10,917 | 10,920     | 10,920 |
>
> **3. Empirical Margins vs. GCN:** We acknowledge that performance gains over GCN are sometimes modest. This is expected, as most benchmarks require only **a few layers** to match 1-WL expressivity, and injectivity is a **sufficient but not necessary** condition for strong performance. Our method is designed to ensure **expressivity and stability at depth**, rather than to optimize margins in shallow regimes. The new experiments demonstrate that our approach achieves consistently strong performance across tasks, without relying on residuals, normalization, or dataset-specific tuning.
>
> **4. Complexity analysis**: We thank the reviewer for pointing out the potential reduction in complexity. While our message passing layers are training-free in theory, in practice we train them when comparing with other GNNs. This is because random weights may inject *noise* into the graph signal, as illustrated in Figures 1b–1c. Consequently, our computational cost remains comparable to standard GNNs like GCN. We did not include a runtime analysis, as we do not expect significant gains. We will clarify this in the paper.
>
> More importantly, our main contributions lie in the theoretical characterization of injective message passing: Proposition 2 and Theorem 1 provide scalable upper bounds on the width for injectivity; Proposition 4 formalizes a topology-aware depth requirement; Lemma 1 gives a stability guarantee for deep injective message passing. We believe these results provide principled insight into scalable and expressive deep GNNs.
>
> [1] Scholkemper, Michael, et al. "Residual Connections and Normalization Can Provably Prevent Oversmoothing in GNNs." ICLR 2025
>
> [2] Luo, Yuankai, Lei Shi, and Xiao-Ming Wu. "Classic gnns are strong baselines: Reassessing gnns for node classification." NeurIPS 2024

---

> > ### Comment · Reviewer_unzM · 2025-08-05
> >
> > I appreciate the author's provided elements in the rebuttal. While I would have hoped to have additional experiments comparing to advanced methods that tackle over-smoothing, it seems that the authors didn't provide them.
> >
> > I have additionally checked the other reviewer's raised comments and weaknesses and I think there are some additional points missing to the current manuscript. In this perspective, I would still keep my "positive" score and keep monitoring the other answers and discussions.

---

### Decision · Program_Chairs · 2025-09-17

**Decision:**

Reject

**Comment:**

This paper claims that the performance degradation in GNNs with increasing depth is due to the lack of injectivity in the commonly employed message passing mechanisms. To deal with this problem, the paper introduces a message passing layer that is provably injective, while it requires no training. A decoupled GNN architecture is also proposed which consists of a small number of the aforementioned layers, followed by a series of feature learning layers.

The reviewers agreed that this paper follows an interesting direction and that the connection between 1-WL expressivity and oversmoothing is worth further investigation. The proposed decoupled GNN architecture appears to be novel. While untrained GNNs have already been explored in the literature, this model is sufficiently different from them. The paper is also well written and easy to read. Most reviewers raised concerns about the experimental evaluation of the proposed model and about the marginal improvements over the baselines. One of the reviewers also argued that many randomly initialized GNNs are likely to exhibit the injectivity property, which limits the novelty of the proposed GNN, while similar theoretical results were presented in [1].

During the rebuttal, the authors conducted further experiments. The proposed model was evaluated on graph classification datasets and on a larger-scale node classification dataset (OGBN-Arxiv). The authors also claimed that the results presented in [1] apply to MPNNs that use sum aggregators, while the results presented in this paper are more general and extend to architectures that use other aggregators.

I personally think that some of the results presented in this paper are very interesting. For instance, Proposition 1 that states that if all distinct node features are linearly independent and $\epsilon$ is irrational, the message passing scheme of Equation (3) is injective is very useful. Also, the bound derived in Proposition 2 for producing linearly independent node features is of high significance. However, I feel that the empirical validation of the proposed decoupled GNN can be further strengthened. Tasks where the structural properties of nodes need to be accurately captured such as graph classification and regression, can benefit from injective message passing mechanisms. The authors evaluated during the rebuttal the method on 5 datasets from the TUDataset collection, but I would suggest they also evaluate it on larger scale datasets such as ogbg-molhiv, ogbg-molpcba and QM9. I also agree with Reviewer 1KdM's concerns  regarding the similarity of some theoretical results with those presented in [1]. In my understanding, the results in this paper are more general and can be applied to other GNN architectures different from the ones in [1], but this needs to be discussed in detail in the paper. Therefore, the paper would greatly benefit from a major revision. Finally, some claims made in the paper seem somewhat misleading. For example, the authors claim that the proposed message passing layer is provably injective but this result holds with high probability over the random initialization of the weight matrices. Based on the above, I am leaning towards rejection and I encourage the authors to address the above issues in the next revision of the paper.

[1] T. Amir et al., Neural Injective Functions for Multisets, Measures and Graphs via a Finite Witness Theorem, NeurIPS 2023.